# QUERY-AWARE HUB PROTOTYPE LEARNING FOR FEW-SHOT 3D POINT CLOUD SEGMENTATION

## ABSTRACT

Few-shot 3D point cloud semantic segmentation (FS-3DSeg) aims to segment novel classes with only a few labeled samples. However, existing metric-based prototype learning methods generate prototypes solely from the support set. This often results in prototype bias, where prototypes overfit support-specific characteristics and fail to generalize to the query distribution, especially in the presence of distribution shifts, which leads to degraded segmentation performance. Although some works make efforts to refine or align these prototypes with queries, prototype bias remains poorly addressed, as the initial prototypes have already deviated significantly from the queries. To address this issue, we propose a novel Query-aware Hub Prototype (QHP) learning method that explicitly models semantic correlations between support and query sets. Specifically, we propose a Hub Prototype Generation (HPG) module that constructs a bipartite graph connecting query and support points, identifies frequently linked support hubs, and generates query-relevant prototypes that better capture cross-set semantics. To further mitigate the influence of bad hubs and ambiguous prototypes near class boundaries, we introduce a Prototype Distribution Optimization (PDO) module, which employs a purity-reweighted contrastive loss to refine prototype representations by pulling bad hubs and outlier prototypes closer to their corresponding class centers. Extensive experiments on S3DIS and ScanNet demonstrate that QHP achieves substantial performance gains over state-of-the-art methods, effectively narrowing the semantic gap between prototypes and query sets in FS-3DSeg.

## 1 INTRODUCTION

Point cloud semantic segmentation assigns semantic labels to each point in a 3D point cloud and is essential for applications like autonomous driving and robotics. Although fully supervised methods (Qi et al., 2017b; Lin et al., 2020; Qian et al., 2022) have made significant progress, they rely heavily on costly manual annotations and struggle to generalize to novel classes. To address these challenges, few-shot 3D point cloud segmentation (FS-3DSeg) has gained increasing attention, aiming to learn generalizable models from abundant base class data and adapt the model to novel classes with only a few labeled point clouds.

Recent FS-3DSeg methods typically adopt metric-based prototype learning frameworks, where prototypes are derived from a few labeled support point clouds, and the unlabeled query set is segmented by measuring similarity between query points and these prototypes. As illustrated in Figure 1 (a)(b), these methods can be broadly categorized into two groups: single-prototype approaches (Mao et al., 2022; He et al., 2023; Liu et al., 2024), which generate global class prototypes via masked average pooling, and multi-prototype methods (Zhao et al., 2021b; An et al., 2024), which enhance prototype diversity through strategies like Farthest Point Sampling (FPS) and local clustering. Ideally, prototypes should serve as semantic bridges between support and query sets, requiring strong alignment with query semantics. However, existing methods generate prototypes solely from the support set, emphasizing internal representativeness or diversity while ignoring semantic relevance to the query. This often causes prototype bias, especially under distribution shifts between support and query sets. For example, intra-class variations (e.g., square vs. round tables) may share limited similarity, causing prototypes to overfit support-specific traits and poorly represent diverse queries. Although some works He et al. (2023); Ning et al. (2023); Hu et al. (2023) leverage query information to refine or align these prototypes in a subsequent stage, prototype bias remains poorly addressed:

Figure 1: Few-shot 3D point cloud semantic segmentation approaches. (a)(b) Previous prototype learning methods generate prototypes solely based on support points. (c) We propose a Query-aware Hub Prototype Learning method that generates prototypes more closely related to query points.

the initial support-only prototypes have already deviated significantly from the query, making subsequent correction challenging. Moreover, uniform sampling strategies like FPS often introduce redundant or query-irrelevant prototypes, further compromising segmentation accuracy. To address these challenges, it is essential to develop a query-aware prototype generation mechanism to narrow the semantic gap between prototypes and the query set and improve segmentation performance.

Realizing the above issues, we propose a **Q**uery-aware **H**ub **P**rototype (**QHP**) learning method, as depicted in Figure 1 (c). Hubs (Radovanović et al., 2009) refer to data points that frequently appear among the nearest neighbors of many other points. Therefore, hubs naturally reflect support-query semantic correlation and are well-suited as prototypes. In recent few-shot image classification studies, some methods (Fei et al., 2021; Trosten et al., 2023; Tang et al., 2025) regard hubs as a nightmare and seek to avoid them, worrying that when a support point is a hub, many query points may be retrieved regardless of their true classes. However, we argue that: **(1) Not all hubs are harmful.** Hubs that emerge within the same class (*i.e.*, good hubs) can capture accurate support-query relationships and serve as effective query-aware prototypes, which helps mitigate prototype bias. **(2) The harmful impact of bad hubs is limited in FS-3DSeg.** Since each support sample contains numerous points and provides richer point-level supervision, each query point's segmentation can be determined by multiple support-query matches, reducing the risk of being misled by any single bad hub. Therefore, instead of suppressing hubs, we leverage them to bridge the semantic gap between support and query, and propose to learn query-aware hub prototypes. Notably, to further mitigate the influence of bad hubs, we optimize their distributions by pulling those near class boundaries closer to corresponding class centers.

The proposed QHP approach introduces two key components: the **H**ub **P**rototype **G**eneration (**HPG**) module and the **P**rototype **D**istribution **O**ptimization (**PDO**) module. Specifically, HPG explicitly models semantic correlations between support and query sets to learn query-relevant hub prototypes. It constructs a bipartite graph connecting query and support points, identifies support hubs with high linking frequency, and performs local clustering around each hub to generate query-relevant prototypes that better capture cross-set semantics. Query segmentation can be conducted by measuring similarities between query points and these hub prototypes. To further mitigate the influence of bad hubs and ambiguous prototypes near class boundaries, we propose a PDO module during training. PDO constructs a global association graph to identify bad hubs, and adopts a purity-reweighted contrastive loss to pull bad hubs and outlier prototypes toward their corresponding class centers. By jointly leveraging the HPG and PDO modules, our QHP facilitates more query-relevant and discriminative prototype learning, effectively narrowing the semantic gap between prototypes and query sets and yielding improved performance in the FS-3DSeg task.

Our main contributions can be summarized as follows:

- We propose a novel Query-aware Hub Prototype (QHP) Learning method for FS-3DSeg, which explicitly models semantic correlations between support and query sets to generate query-relevant prototypes, addressing prototype bias and narrowing the semantic gap.
- We propose a Hub Prototype Generation (HPG) module to identify support hubs and generate query-relevant hub prototypes that better capture cross-set semantics.

- We design a Prototype Distribution Optimization (PDO) module, optimizing the distributions of bad hubs and outlier prototypes via a purity-reweighted contrastive loss.
- Extensive experiments on S3DIS and ScanNet demonstrate that QHP achieves state-of-the-art performance.

## 2 RELATED WORK

### 2.1 3D POINT CLOUD SEMANTIC SEGMENTATION

Recent 3D point cloud segmentation methods can be broadly categorized into MLP-based (*e.g.*, PointNet (Qi et al., 2017a) and RandLA-Net (Hu et al., 2020)), convolution-based (*e.g.*, PointCNN (Li et al., 2018), KPConv (Thomas et al., 2019), and RandLA-Net (Hu et al., 2020)), and Transformer-based approaches such as Point Transformer (Zhao et al., 2021a) and Point Transformer V2 (Wu et al., 2022). Although these methods demonstrate strong performance through local feature aggregation or self-attention mechanisms, they typically require expensive, large-scale annotations and struggle to generalize to novel classes unseen during training.

### 2.2 FEW-SHOT 3D POINT CLOUD SEGMENTATION

Recent FS-3DSeg methods primarily adopt the prototype-based paradigm built upon metric learning. These methods can be broadly categorized into single-prototype and multi-prototype approaches. Single-prototype methods summarize each class using a single representative prototype from the support set. For instance, ProtoNet defines the prototype as the class-wise mean of support features. To mitigate distribution shifts between support and query sets, 2CBR (Zhu et al., 2023) explicitly models such biases, and DPA (Liu et al., 2024) employs query-guided attention to generate task-adaptive prototypes. Seg-NN/PN (Zhu et al., 2024) designs a lightweight module to optimize support-query interaction for prototype generation. However, these methods lack prototype diversity and are unsuitable for handling complex data. To capture intra-class variations, multi-prototype approaches generate multiple prototypes per class. AttMPTI (Zhao et al., 2021b) employs farthest point sampling (FPS) to extract diverse local prototypes. Stratified Transformer (Lai et al., 2022) combines hierarchical sampling with cross-window self-attention. COSeg (An et al., 2024) maintains a momentum-updated pool of base class prototypes. Besides, recent generalized FS-3DSeg methods An et al. (2025a) leverage vision-language models. Despite these advances, most methods rely solely on support data to generate prototypes, yielding prototypes biased toward support distribution and poorly aligned with queries, thus limiting generalization to novel classes.

### 2.3 THE HUBNESS PHENOMENON AND HUBS

Hubness (Radovanovic et al., 2010; Radovanović et al., 2009) describes the tendency of certain points, called *hubs*, to appear frequently in nearest-neighbor lists. It has been studied in areas like multi-view clustering (Xu et al., 2025) and cross-modal retrieval (Bogolin et al., 2022; Wang et al., 2023). In few-/zero-shot classification tasks, prior works (Dinu & Baroni, 2015; Xiao et al., 2021; Cheraghian et al., 2019; Trosten et al., 2023) mostly view hubs as harmful, as query points may be misclassified when dominated by support hubs from different classes. In contrast, we argue that good hubs are beneficial and are primary in our scenario. We thus exploit hubs via query-aware hub prototype learning and mitigate bad hub distance optimization to narrow query-support gaps.

## 3 METHOD

### 3.1 PROBLEM FORMULATION AND OVERVIEW

**Problem Formulation.** FS-3DSeg aims to predict per-point semantic labels for query point clouds using a few labeled support samples. Episodic learning (Zhao et al., 2021b) is typically employed to simulate the few-shot learning process, where each $C$-way $K$-shot episode includes a support set $S = \{(I_s^{c,k}, M_s^{c,k})_{k=1}^K\}_{c=1}^C$ and a query set $Q = \{(I_q^l, M_q^l)\}_{l=1}^L$. Each point cloud $I_s^{c,k}, I_q^l \in \mathbb{R}^{T \times (3+f_0)}$ contains $T$ points, each represented by 3D coordinates and auxiliary features (*e.g.*, RGB). $M_s^{c,k} \in \{0,1\}^T$ denotes the binary ground truth (GT) mask indicating whether each point in $I_s^{c,k}$

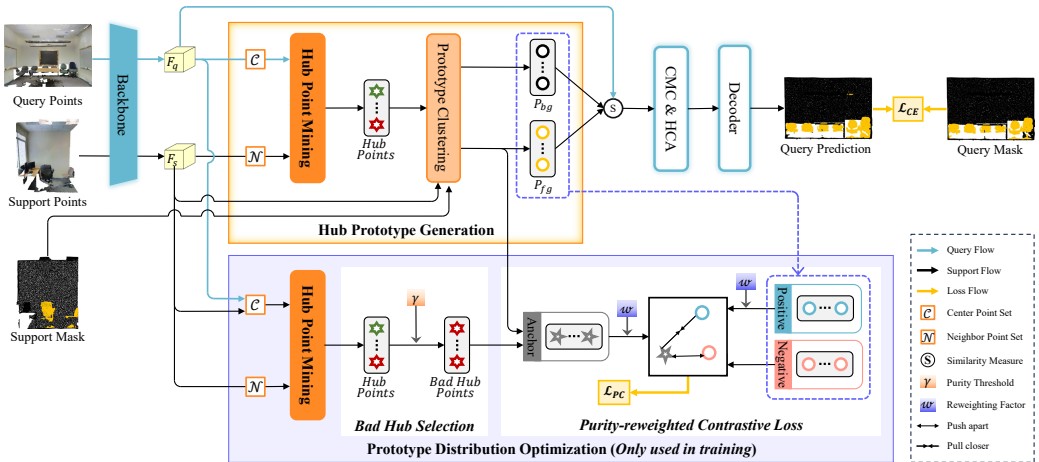

Figure 2: The framework of our Query-aware Hub Prototype Learning method. Initially, we design an HPG module to select support hubs and generate query-relevant hub prototypes. Moreover, during training, a PDO module is integrated to optimize the distribution of bad hubs and outlier prototypes. For clarity, we present the model under the 1-way 1-shot setting.

belongs to class $c$, while $M_q^l$ denotes GT labels for query point cloud $I_q^l$. During inference, our goal is to predict the query labels $\hat{M}_q$ for the query points in $I_q$ under the guidance of the support set $S$.

**Overview.** Figure 2 illustrates the architecture of the proposed QHP framework, comprising two key components: a Hub Prototype Generation (HPG) module and a Prototype Distribution Optimization (PDO) module. We first use a shared backbone $\Phi$ to extract point-wise features from the support and query point clouds: $F_s = \Phi(I_s) \in \mathbb{R}^{C \times K \times T \times D}$ and $F_q = \Phi(I_q) \in \mathbb{R}^{L \times T \times D}$, where $D$ is the channel dimension. In the HPG module, a Hub Point Mining (HPM) module identifies hub points from $S$, which are used to generate hub prototypes $P$ via local clustering. These prototypes are matched with query features $F_q$ through similarity measures, and further refined via the CMC and HCA modules (An et al., 2024) to yield query predictions $\hat{M}_q$. To mitigate the influence of bad hubs and ambiguous prototypes, our PDO module identifies bad hubs by thresholding their purity and applies a Purity-reweighted Contrastive (PC) loss to promote intra-class compactness. During training, our model is jointly optimized by a cross-entropy (CE) loss and the proposed PC loss.

Subsequently, we provide a detailed description of the HPG module, PDO module and each loss.

## 3.2 HUB PROTOTYPE GENERATION

To mitigate prototype bias, we propose an HPG module. It first identifies frequently occurring support hubs via a Hub Point Mining (HPM) module, then generates query-relevant hub prototypes through Hub Prototype Clustering.

**Hub Point Mining.** HPM identifies hub points through three sequential steps, as illustrated in Figure 3(a)–(c).

*Step 1: $k$-Nearest Neighbor Mining.* Given a center point set $\mathcal{C}$ and a neighbor point set $\mathcal{N}$, a bipartite graph is constructed by connecting each center point $c \in \mathcal{C}$ to its $k$-nearest ($k$NN) neighbors in $\mathcal{N}$ via cosine similarity measure. The $k$-nearest neighbors of $c$ are formulated as $k\text{NN}(c, \mathcal{N})$.

*Step 2: Hubness Score Statistic.* The hubness score $s(n)$ quantifies how frequently a point $n \in \mathcal{N}$ is selected as neighbors by all center points in $\mathcal{C}$, defined as:

$$s(n) = \sum_{c \in \mathcal{C}} \mathbb{1}\left(n \in k\text{NN}(c, \mathcal{N})\right) + \varepsilon, \tag{1}$$

where $\mathbb{1}(\cdot)$ denotes the Iverson bracket indicator function, returning 1 if the condition holds and 0 otherwise. A small positive constant $\varepsilon$ is added to avoid zero scores caused by potential outliers, ensuring $s(n) > 0$ for all $n$. The collective hubness scores for all points in $\mathcal{N}$ are denoted as $s(\mathcal{N})$.

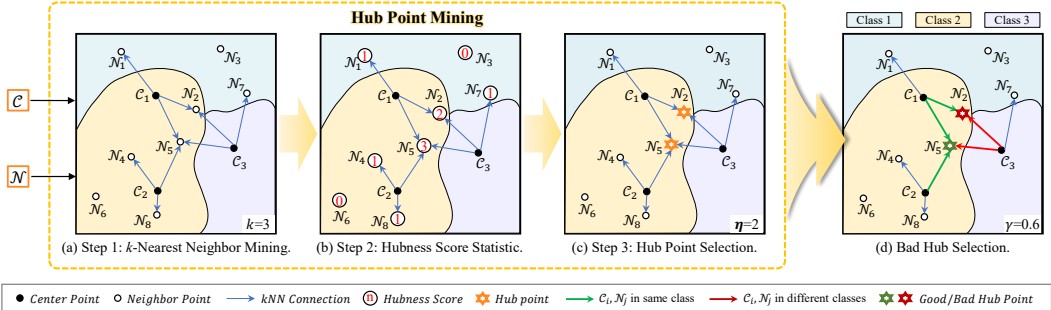

Figure 3: Illustration of Hub Point Mining and Bad Hub Selection modules. We give an example under hyperparameters $k$=3, $\eta$=2, and $\gamma$=0.6. (a)–(c) Hub Point Mining: using center points $\{\mathcal{C}_1, \mathcal{C}_2, \mathcal{C}_3\}$ and neighbor points $\{\mathcal{N}_1, \ldots, \mathcal{N}_8\}$ as input, a $k$NN graph is constructed with $k$ =3. After calculating hubness scores, the top $\eta$=2 points with the highest hubness scores are selected as hubs. (d) Bad Hub Selection: Hubs with purity below the threshold $\gamma$ =0.6 are selected as bad hubs.

*Step 3: Hub Point Selection.*

To identify nodes most frequently regarded as neighbors by center points, we select a subset $\mathcal{H} \subseteq \mathcal{N}$ consisting of the top-$\eta$ neighbor points with the highest hubness scores, defined as:

$$\mathcal{H} = \{n \in \mathcal{N} \mid s(n) \in \text{Top-}\eta(s(\mathcal{N}))\}. \tag{2}$$

**Hub Prototype Clustering.** Before prototype clustering, within the HPM module, we treat query points as exclusive center nodes $\mathcal{C}$ (prioritizing query-driven selection of support hubs highly relevant to queries) and support points as neighbor nodes $\mathcal{N}$, then select Top-$\eta$ hub points for each class to construct hub point set $\mathcal{H}$.

A prototype set $P = P_{fg} \cup P_{bg}$, where $P_{fg}$ and $P_{bg}$ denote foreground/background prototypes, is generated by conducting point-to-seed clustering (Zhao et al., 2021b) on support features localized around each hub point, defined as:

$$
\begin{aligned}
P_{fg} &= \mathcal{F}_{clus}\left(F_s \odot M_s, \mathcal{H}_{fg}\right), \mathcal{H}_{fg} = \mathcal{H} \odot M_s, \\
P_{bg} &= \mathcal{F}_{clus}\left(F_s \odot \neg M_s, \mathcal{H}_{bg}\right), \mathcal{H}_{bg} = \mathcal{H} \odot \neg M_s,
\end{aligned}
\tag{3}
$$

where $\odot$ denotes the Hadamard product; $M_s$ and $\neg M_s$ are the GT mask and its inverse for support set; $\mathcal{H}_{fg}$ and $\mathcal{H}_{bg}$ are foreground/background hub point subsets; and $\mathcal{F}_{clus}$ denotes the clustering operation. After that, we obtain $\eta$ prototypes per class, yielding a total of $(C + 1) \cdot \eta$ prototypes.

Notably, although support hubs $\mathcal{H}$ originate from $S$, they are geometrically closer to points in $Q$ as they retain only those support points that best match the query distribution. Consequently, the derived hub prototypes are more aligned with $Q$ in the metric space, facilitating improved prototype-query matching and enhanced segmentation performance.

### 3.3 PROTOTYPE DISTRIBUTION OPTIMIZATION

In the PDO module, we select potential bad hubs, and then adopt a Purity-reweighted Contrastive (PC) loss to suppress these bad hubs and optimize the prototype distribution.

**Bad Hub Selection.** To select bad hubs, which occur both at query-support boundaries and inter-class boundaries within the support set, we merge $Q$ and $S$ as the center point set, treat $S$ as the neighbor set $\mathcal{N}$, construct a global association graph via $k$-NN, and apply the HPM module to identify hub points $\mathcal{H}$ from $S$.

After that, we identify all potential bad hubs within $\mathcal{H}$ that have stronger connections to center points belonging to different classes, as shown in Figure 3(d). This process involves three steps:

First, we compute the number of times each hub $h$ is connected to center points of the same class, denoted as $t(h)$:

$$t(h) = \sum_{c \in \mathcal{C}} \mathbb{1}\left(h \in k\text{NN}(c, \mathcal{N})\right) \cdot \mathbb{1}\left(M_h = M_c\right), \tag{4}$$

where $M_c$, $M_h$ are class labels of $c$ and $h$, respectively.

Next, a purity $\mathscr{P}(h)$ is defined to represent the proportion of connections to center points of the same class, given by:

$$\mathscr{P}(h) = t(h)/s(h). \tag{5}$$

Finally, the bad hub point set $\mathcal{BH}$ is filtered out using a purity threshold $\gamma \in (0, 1)$, formulated as:

$$\mathcal{BH} = \{h \in \mathcal{H} \mid \mathscr{P}(h) < \gamma\}. \tag{6}$$

**Purity-reweighted Contrastive Loss.** To further eliminate the influence of bad hubs and outlier prototypes, we aim to pull them back to their cluster centers. A typical solution is contrastive learning (Wang & Liu, 2021), which has been widely adopted in various areas to pull positive pairs closer and push negative pairs apart. Despite these successes, contrastive loss has not been explored in FS-3DSeg. However, directly applying standard contrastive loss to bad hub anchors is suboptimal: low-purity anchors, which have high similarity to samples from other classes, tend to lie far from their true class centers and are more likely to be confused, thus requiring stronger guidance to be correctly aligned. To tackle this issue, we propose a Purity-reweighted Contrastive (PC) loss, which dynamically adjusts the attraction strength based on the purity of each anchor sample.

We first introduce a purity reweighting factor $w(a)$ to quantify the strength with which anchor $a \in \mathcal{A} = \{P_{fg} \cup \mathcal{BH}\}$ is pulled toward positive prototypes, formulated as:

$$w(a) = \begin{cases} 1 - \mathscr{P}(a) & \text{if} \quad a \in \mathcal{BH} \\ 1 & \text{otherwise} \quad a \in P_{fg} \end{cases}, \tag{7}$$

where the first line assigns a higher weight ($w(a) \rightarrow 1$) inversely proportional to purity for bad hub anchors ($a \in \mathcal{BH}$) with low purity ($\mathscr{P}(a) \rightarrow 0$), strongly pulling them toward class centers; the second line assigns a fixed weight ($w(a) = 1$) for all foreground prototype anchors ($a \in P_{fg}$), ensuring intra-class compactness and inter-class discriminability.

Based on the defined purity reweighting factor $w(a)$, we formulate our purity-reweighted contrastive loss for all anchors $\mathcal{A}$ as follows,

$$\mathcal{L}_{\text{PC}} = -\frac{1}{|\mathcal{A}|} \sum_{a \in \mathcal{A}} \log \frac{w(a) \cdot \sum_{p \in P^+(a)} \exp(\text{sim}(a,p)/\tau)}{w(a) \cdot \sum_{p \in P^+(a)} \exp(\text{sim}(a,p)/\tau) + \sum_{p \in P^-(a)} \exp(\text{sim}(a,p)/\tau)}, \tag{8}$$

where $P^+(a) = \{p \in P \mid M_p = M_a\}$ and $P^-(a) = \{p \in P \mid M_p \neq M_a\}$ are positive set and negative set, respectively; $\tau$ is a temperature parameter that controls smoothing; $\text{sim}(a,p)$ is the similarity between $a$ and $p$.

## 3.4 TOTAL LOSS

During training, the proposed model is supervised by two loss functions, *i.e.*, a standard cross-entropy loss $\mathcal{L}_{\text{CE}}$ that serves to optimize segmentation results, and the proposed PC loss $\mathcal{L}_{\text{PC}}$ to optimize the prototype distributions. Specifically, the cross-entropy loss $\mathcal{L}_{\text{CE}}$ is defined as:

$$\mathcal{L}_{\text{CE}} = -\frac{1}{L} \sum_{l=1}^{L} M_q^l \log(\hat{M}_q^l), \tag{9}$$

where $M_q^l$ and $\hat{M}_q^l$ represent GT mask and the predicted mask for query sample.

Overall loss is a weighted combination of $\mathcal{L}_{\text{CE}}$ and $\mathcal{L}_{\text{PC}}$ with a balancing weight $\lambda$, represented as:

$$\mathcal{L}_{\text{total}} = \mathcal{L}_{\text{CE}} + \lambda \cdot \mathcal{L}_{\text{PC}}. \tag{10}$$

## 4 EXPERIMENTS

### 4.1 IMPLEMENTATION DETAILS

**Training.** Experiments were conducted on the S3DIS (Armeni et al., 2016) and ScanNet (Dai et al., 2017) datasets. Data preprocessing follows An et al. (2024): rooms are divided into 1m

| Methods | 1-way 1-shot | | | 1-way 5-shot | | | 2-way 1-shot | | | 2-way 5-shot | | |
|---|---|---|---|---|---|---|---|---|---|---|---|---|
| | $S^0$ | $S^1$ | mean | $S^0$ | $S^1$ | mean | $S^0$ | $S^1$ | mean | $S^0$ | $S^1$ | mean |
| AttMPTI (Zhao et al., 2021b) | 36.32 | 38.36 | 37.34 | 46.71 | 42.70 | 44.71 | 31.09 | 29.62 | 30.36 | 39.53 | 32.62 | 36.08 |
| QGE (Ning et al., 2023) | 41.69 | 39.09 | 40.39 | 50.59 | 46.41 | 48.50 | 33.45 | 30.95 | 32.20 | 40.53 | 36.13 | 38.33 |
| QGPA (He et al., 2023) | 35.50 | 35.83 | 35.67 | 38.07 | 39.70 | 38.89 | 25.52 | 26.26 | 25.89 | 30.22 | 32.41 | 31.31 |
| Seg-PN$^\dagger$ (Zhu et al., 2024) | 37.01 | 40.43 | 38.72 | 39.72 | 43.02 | 41.37 | 33.21 | 37.02 | 35.12 | 39.08 | 39.16 | 39.12 |
| COSeg (An et al., 2024) | 46.31 | 48.10 | 47.21 | 51.40 | 48.68 | 50.04 | 37.44 | 36.45 | 36.95 | 42.27 | 38.45 | 40.36 |
| COSeg$^\dagger$ | 45.93 | 47.48 | 46.71 | 48.47 | 48.72 | 48.60 | 37.17 | 37.03 | 37.10 | 41.65 | 38.38 | 40.02 |
| **QHP (ours)** | **50.33** | **48.73** | **49.53** | **52.27** | **49.64** | **50.96** | **38.86** | **37.84** | **38.35** | **43.90** | **40.04** | **41.97** |

Table 1: Comparison of mIoU (%) performance between our method and previous FS-3DSeg approaches on the **S3DIS** dataset. Methods with $^\dagger$ are re-implementation using their official code. The best results are highlighted in **bold**, and the second-best results are marked in blue.

| Methods | 1-way 1-shot | | | 1-way 5-shot | | | 2-way 1-shot | | | 2-way 5-shot | | |
|---|---|---|---|---|---|---|---|---|---|---|---|---|
| | $S^0$ | $S^1$ | mean | $S^0$ | $S^1$ | mean | $S^0$ | $S^1$ | mean | $S^0$ | $S^1$ | mean |
| AttMPTI (Zhao et al., 2021b) | 34.03 | 30.97 | 32.50 | 39.09 | 37.15 | 38.12 | 25.99 | 23.88 | 24.94 | 30.41 | 27.35 | 28.88 |
| QGE (Ning et al., 2023) | 37.38 | 33.02 | 35.20 | 45.08 | 41.89 | 43.49 | 26.85 | 25.17 | 26.01 | 28.35 | 31.49 | 29.92 |
| QGPA (He et al., 2023) | 34.57 | 33.37 | 33.97 | 41.22 | 38.65 | 39.94 | 21.86 | 21.47 | 21.67 | 30.67 | 27.69 | 29.18 |
| Seg-PN$^\dagger$ (Zhu et al., 2024) | 33.98 | 29.45 | 31.72 | 37.24 | 31.78 | 34.51 | 28.20 | 26.72 | 27.46 | 35.52 | 30.40 | 32.96 |
| COSeg (An et al., 2024) | **41.73** | 41.82 | 41.78 | 48.31 | 44.11 | 46.21 | **28.72** | 28.83 | **28.78** | 35.97 | 33.39 | 34.68 |
| COSeg$^\dagger$ | 40.57 | 41.94 | 41.26 | 49.43 | 43.57 | 46.50 | 28.06 | 28.92 | 28.49 | 35.49 | 34.03 | 35.06 |
| **QHP (ours)** | 40.70 | **42.92** | **41.81** | 50.10 | 44.80 | 47.45 | 28.45 | 29.07 | 28.76 | **36.11** | **34.30** | **35.21** |

Table 2: Comparison of mIoU (%) performance between our method and previous FS-3DSeg approaches on the **ScanNet** dataset. Methods with $^\dagger$ are re-implementation using their official code. The best results are highlighted in **bold**, and the second-best results are marked in blue.

$\times$ 1m blocks, input points grid-sampled at 0.02m intervals, and 20,480 points are selected after voxelization to standardize input size. Data augmentation and backbone pre-training were applied as in An et al. (2024), with each fold pre-trained for 100 epochs. Meta-training was performed over 40,000 episodes using AdamW with a learning rate of $5 \times 10^{-5}$ and weight decay of 0.01. For testing, 1,000 episodes per class were sampled in 1-way settings, and 100 episodes per combination in 2-way settings. We used 100 prototypes per class ($\eta = 100$); in $k$-shot settings ($k > 1$), $\eta/k$ prototypes were selected from each shot and concatenated to form the final prototypes.

**Parameters.** In both the HPG and PDO modules, the number of neighbors for hub point mining is set to $k = 5$. In the HPG module, the number of hub points/prototypes per class is set to $\eta = 100$. In the PDO module, the bad hub purity threshold is set to $\gamma = 0.6$ in (Eq. 6). In the total loss function (Eq. 10), the balance weight for $\mathcal{L}_{PC}$ is set to $\lambda = 0.1$.

## 4.2 COMPARISON RESULTS

**Comparison with Previous Methods.** We compare our QHP with prior works including AttMPTI (Zhao et al., 2021b), QGE (Ning et al., 2023), QGPA (He et al., 2023), Seg-PN (Zhu et al., 2024) and COSeg (An et al., 2024) on S3DIS (Armeni et al., 2016) and ScanNet (Dai et al., 2017) datasets. Additionally, we exclude MM-FSS (An et al., 2025b) method as it uses multimodal information, yielding an unfair comparison. The Seg-PN results presented in Table 1 and Table 2 are reproduced based on the corrected few-shot setting proposed by COSeg (An et al., 2024), and the Seg-NN results in Table 4 of Section 4.3 are obtained using the same setting.

• **S3DIS.** Table 1 shows that our QHP consistently outperforms prior approaches across all settings. Specifically, compared to the baseline method COSeg$^\dagger$, QHP achieves performance gains of 2.82% and 2.36% in the 1-way settings, and enhancements of 1.25% and 1.95% in the 2-way settings. These gains can be attributed to the hub prototypes generated by our method. Unlike COSeg$^\dagger$, which relies on FPS-based prototype generation that may produce redundant or irrelevant prototypes, HPG module effectively identifies hub points to generate query-relevant prototypes. Additionally, opti-

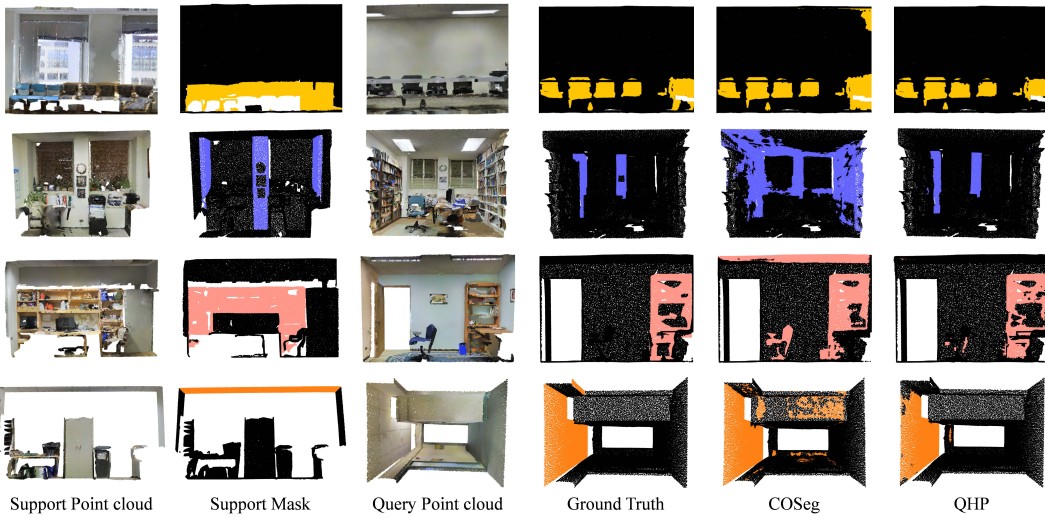

Support Point cloud    Support Mask    Query Point cloud    Ground Truth    COSeg    QHP

Figure 4: Qualitative comparisons between our proposed model QHP and COSeg. Each row, from top to bottom, represents the 1-way 1-shot task with the target category as chair (yellow), column (blue), bookcase (pink), and ceiling (orange), respectively.

| Baseline | HPG | PDO | mIoU (%) |
|:---:|:---:|:---:|:---:|
| ✓ | | | 45.93 |
| ✓ | ✓ | | 47.37 |
| ✓ | ✓ | ✓ | **50.33** |

Table 3: Ablation study of key components in QHP method.

| Methods | $S^0$ | $S^1$ | mean |
|:---|:---:|:---:|:---:|
| COSeg[†] | 45.93 | 47.48 | 46.71 |
| **Ours** | **47.37** | **48.12** | **47.73** |
| Seg-NN[†] | 25.86 | 30.54 | 28.20 |
| **Ours** | **28.28** | **30.99** | **29.64** |

Table 4: Performance evaluation under different baselines (1-way 1-shot setting).

mizing prototype distributions further contributes to the superior performance of our model. When compared to query-guided methods such as QGE and QGPA, QHP demonstrates more significant advantages in the 1-way tasks, with improvements of $9.14\%$ and $2.46\%$, respectively. This highlights the superiority of our method in enhancing the discriminability of prototypes.

• **ScanNet.** Table 2 shows that QHP outperforms all prior methods across all settings, further validating the effectiveness and applicability of our approach. Notably, in the 1-way 5-shot task, QHP achieves a mIoU of $47.45\%$ and outperforms COSeg[†] by $0.95\%$, which highlights QHP's adaptability to the complex ScanNet dataset. We note that our performance gains are more pronounced in 5-shot than 1-shot settings: with more support samples available, QHP can mine important hub points from a larger pool of support points to generate query-relevant prototypes. However, improvements on ScanNet are less substantial than S3DIS, as the dataset's higher complexity and inter-class overlap pose greater challenges for distinguishing similar categories.

**Qualitative Results.** In Figure 4, we compare the results from our QHP (6th column) with COSeg (5th column). QHP improves object boundaries and category shapes, especially for column contours (blue, 2nd row), capturing finer details and reducing redundancy. PDO module excels in chair class (yellow, 1st row), resolving boundary ambiguities for more precise segmentation. Overall, QHP delivers cleaner, more accurate results with improved boundary delineation and reduced redundancy.

### 4.3 ABLATION STUDY

We present an ablation study on the S3DIS dataset under 1-way 1-shot $S^0$ setting to validate the effectiveness of HPG and PDO modules, as well as hyperparameter settings.

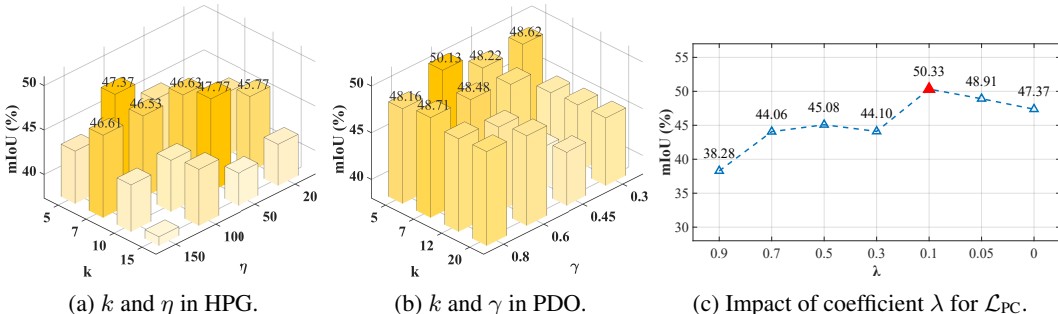

(a) $k$ and $\eta$ in HPG.  (b) $k$ and $\gamma$ in PDO.  (c) Impact of coefficient $\lambda$ for $\mathcal{L}_{PC}$.

Figure 5: Sensitivity analysis of parameters in HPG, PDO and the coefficient $\lambda$ for $\mathcal{L}_{PC}$.

| Hub Ratio | 0% | 30% | 50% | 80% | 100% |
|---|---|---|---|---|---|
| mIoU (%) | 45.93 | 42.93 | 42.32 | 45.47 | **47.37** |

Table 5: Analyze the ratio of hub prototypes to total prototypes in HPG.

| Diff. Losses | Contrastive Loss | Our PC Loss |
|---|---|---|
| mIoU (%) | 49.82 | **50.33** |

Table 6: Comparison between PC loss and contrastive loss.

**Effects of Core Components.** Using COSeg as the baseline, we conduct experiments to evaluate the effectiveness of the two core components, *i.e.*, the HPG and PDO modules. As shown in Table 3, incorporating the HPG module improves the mIoU from $45.93\%$ to $47.37\%$ ($+1.44\%$), while the addition of the PDO module further increases the mIoU to $50.33\%$ ($+2.96\%$), showing that the joint use of HPG and PDO significantly enhances the query relevance and discriminability of the prototypes, thereby leading to substantial performance gains.

**Evaluation on Additional Baselines.** To demonstrate the generalization capability of our method, we apply it to additional baselines, including SegNN Zhu et al. (2024), as shown in Table 4. We extend SegNN's single-prototype approach to a multi-prototype variant by replacing it with a method that employs FPS and local clustering. Since SegNN is a non-parametric baseline that does not use loss functions for training, we only apply our HPG module. The results show that our method outperforms Seg-NN, demonstrating the generalization capability of our approach.

**Effects of Hub Prototypes from HPG.** Without using the PDO module, we fix the number of prototypes to 100 per class and mix FPS-based prototypes with our hub prototypes in varying ratios. As shown in Table 5, as the hub prototype ratio increases from $0\%$ (*i.e.*, baseline COSeg) to $50\%$, performance slightly drops, likely because prototype class diversity is reduced while the query relevance of prototypes remains limited, causing suboptimal performance. Beyond $50\%$, performance steadily improves, peaking at $100\%$, demonstrating that hub prototypes better capture support-query semantic correlation and provide more discriminative representations, significantly enhancing segmentation.

**Impact of Parameters $k$ and $\eta$ in HPG.** We analyze the impact of the number of neighborhoods $k$ in Eq. 1 and the number of hub points $\eta$ in Eq. 2, as shown in Figure 5a. The best performance is achieved when $k = 12$ and $\eta = 50$, followed by $k = 5$ and $\eta = 100$. For fair comparison with other multi-prototype methods with 100 prototypes, we select the setting $k = 5$ and $\eta = 100$.

**Impact of Parameters $k$ and $\gamma$ in PDO.** We analyze the impact of the number of neighborhood $k$ and purity threshold $\gamma$ in PDO, as shown in Figure 5b. The best performance is achieved when $k = 5$ and $\gamma = 0.6$; thus, these parameters are selected in our setup.

**Effects of Different Contrastive Losses in PDO.** To verify the superiority of our proposed PC loss, we compared it with the standard contrastive loss in the PDO module, as shown in Table 6. The PC loss yields a $0.51\%$ performance improvement, demonstrating that the reweighting factor in our PC loss more effectively pulls outlier prototypes and bad hubs near cluster boundaries toward their respective class centers, reducing boundary ambiguity and enhancing overall performance.

**Impact of Coefficient $\lambda$ in Total Loss.** Figure 5c illustrates the impact of the weight $\lambda$ of the PC loss in Eq. 10. Value of $\lambda = 0.1$ yields the best results, indicating that the model achieves a balance

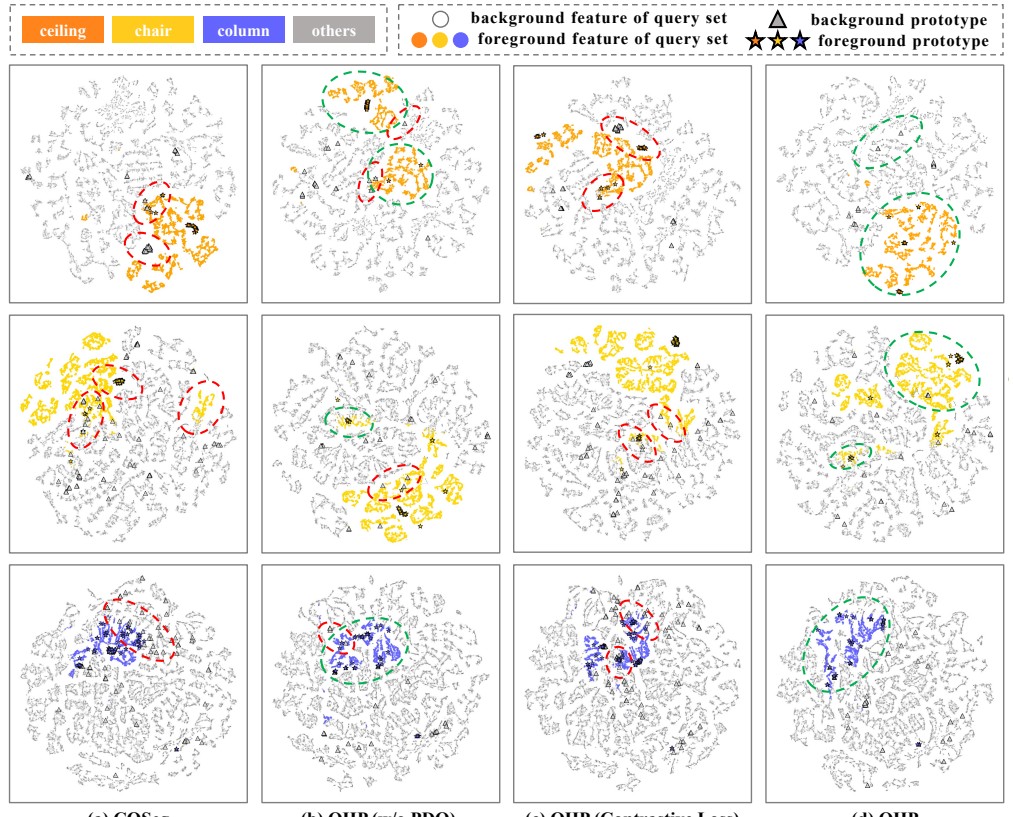

Figure 6: Qualitative comparison of query feature distribution and support prototype distribution under the 1-way 1-shot $S^0$ setting on S3DIS. Red circles highlight prototypes with center deviation or boundary ambiguity, while green circles show the improved distributions.

between class boundary distribution and query segmentation. However, continued increases in $\lambda$ could pull prototypes and hub points too tightly into clusters, thereby harming model performance.

**Comparison of query feature distribution and prototype distribution.** We visualize the feature space for multiple test episodes via t-SNE Van der Maaten & Hinton (2008), as shown in Figure 6. These visualizations include foreground/background features from the query set, as well as foreground/background support prototypes. (a) For COSeg, support-only prototypes often deviate from the true query distribution—showing both class-center shifts and boundary confusion. (b) After introducing our HPG module, class-center misalignment is noticeably reduced, but ambiguous regions near decision boundaries remain. (c) While retaining the HPG module, we replace our PC loss with standard contrastive loss in PDO module. Results indicate that although separability is slightly improved and boundary ambiguity is partially alleviated, residual confusion near decision boundaries remains unresolved. (d) With our QHP equiped with PC loss, boundary issues are further corrected, and prototypes become more compact and better aligned with their corresponding query clusters.

## 5 CONCLUSION

We propose a Query-aware Hub Prototype (QHP) framework for few-shot 3D point cloud semantic segmentation, addressing limitations of prior methods relying solely on support prototypes. QHP models semantic correlations between support and query sets to enhance prototype relevance, with two key modules: Hub Prototype Generation (HPG) module, identifying high-frequency hub points from support to generate query-relevant prototypes; and Prototype Distribution Optimization (PDO) module, reducing the impact of bad hubs and ambiguous prototypes via purity-reweighted contrastive loss. Experiments on S3DIS and ScanNet show the superiority of the proposed model.

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

# A APPENDIX

In the Appendix, we provide additional experimental details and analyses to complement the main paper. Sections A.1 and A.2 present further ablation studies on the S3DIS dataset, while Section A.3 reports both the overall and detailed computation costs. Section A.4 performs ablation experiments on the ScanNet dataset to evaluate the robustness of our model hyperparameters across different datasets. More visualization results and failure cases are provided in Section A.5. Finally, to assess the effectiveness of our method under significant domain shifts, we extends our approach to cross-domain scenarios (S3DIS → ScanNet) in Section A.6.

Notably, we used large language models (LLMs) solely to aid in drafting and polishing the writing of this paper. All scientific content, experimental design, and results are original to this work.

## A.1 MORE EXPERIMENT DETAILS

We conducted experiments on the S3DIS and ScanNet datasets. The S3DIS dataset Armeni et al. (2016) comprises five large-scale indoor areas across three buildings, annotated with 12 semantic classes for segmentation tasks. The ScanNet dataset Dai et al. (2017) contains 1,513 point cloud scans from 707 indoor scenes, covering 20 semantic categories. Compared to S3DIS, ScanNet features more irregular point clouds, rendering segmentation more challenging. Following An et al. (2024), we also adopt the first three blocks of the Stratified Transformer (Lai et al., 2022) as the backbone of our model. For the S3DIS and ScanNet datasets, the input features consist of both XYZ coordinates and RGB colors. All settings are implemented in PyTorch. The 1-way and 2-way 1-shot settings are trained on 4 RTX A4000 GPUs, while the 2-way 5-shot setting is trained on 4 RTX 3090 GPUs.

## A.2 MORE ABLATION STUDIES ON S3DIS DATASET

**Analysis of hyperparameters $k, \eta$ in HPG and $k, \gamma$ in PDO**. In Section 4.3, we have conduct extensive ablation studies on the parameters. The detailed results of the parameter analysis are presented in Table 7 and Table 8, with specific sections illustrated in Figure 5a and Figure 5b.

| mIoU (%) | 20 | 50 | 100 | 150 |
|---|---|---|---|---|
| **5** | 41.71 | 44.52 | 47.37 | 43.24 |
| **7** | 44.41 | 46.63 | 46.53 | 46.61 |
| **10** | 45.77 | **47.77** | 45.77 | 42.60 |
| **15** | 41.98 | 40.98 | 43.67 | 38.34 |

| mIoU (%) | 0.3 | 0.45 | 0.6 | 0.8 |
|---|---|---|---|---|
| **5** | 48.62 | 48.22 | **50.13** | 48.16 |
| **7** | 44.93 | 47.97 | 48.48 | 48.71 |
| **12** | 45.13 | 42.52 | 47.05 | 47.95 |
| **20** | 45.24 | 43.69 | 47.62 | 48.07 |

Table 7: Experimental Results on Parameter Sensitivity of HPG (column: $k$, row: $\eta$).

Table 8: Experimental Results on Parameter Sensitivity of PDO (column: $k$, row: $\gamma$).

**Impact of Different Hubs in HPG**. In the HPG module, to verify the impact of good/bad hubs on hub prototype quality, we use only good hubs versus all hubs (good + bad) in prototype clustering, as in Table 9. Results show that performance with prototypes from only good hubs is suboptimal compared to all hubs. We analyze that prototypes derived solely from good hubs, while effective in reducing noise interference during testing, limit training by capturing only intra-class semantics, weakening class discrimination. In contrast, incorporating bad hubs covers overlooked inter-class discriminative information from class boundaries and complex scenarios, boosting training by refining the model's discriminative ability and yielding a 3.04% improvement. Notably, when combined with the PDO module, bad hub distribution is further optimized, avoiding interference from excessive outliers during testing.

**Impact of Different Anchors in PC Loss**. In PC loss, using both foreground prototypes and bad hubs as anchors improves performance by 1.75% compared to using only foreground prototypes (Table 10). Our analysis reveals that relying solely on foreground prototypes as anchors prioritizes intra-class compactness but underemphasizes semantic associations in boundary regions. Combining foreground prototypes and good hubs as anchors degrades performance, as good hubs already

| Types of Hubs | Good hubs | All hubs |
|:---:|:---:|:---:|
| mIoU (%) | 44.33 | **47.37** |

Table 9: Impact of different hubs in HPG.

have strong intra-class compactness, and using them as anchors overly reinforces this, squeezing beneficial variance, blurring fine-grained distinctions and boundaries. In contrast, introducing bad hub anchors enables PC loss to further emphasize boundaries/outliers, pull bad hubs to class centers, accommodate complex scenario ambiguities, break isolation between outlier prototypes and bad hubs, and boost cross-set alignment.

| Types of Anchors | $P_{fg}$ | $P_{fg}$ & Good hubs | $P_{fg}$ & Bad hubs |
|:---:|:---:|:---:|:---:|
| mIoU (%) | 48.58 | 47.83 | **50.33** |

Table 10: Impact of different anchors in PC Loss.

**Impact of Temperature Parameter $\tau$ in PC Loss**. We also conduct experiments to analyze the effects of the temperature parameter ($\tau$) in Equation 8. The ablation values of ($\tau$) cover both typical settings commonly used in contrastive learning (e.g., $\tau = 0.02$ in Wang et al. (2024), $\tau = 0.1$ in Chen et al. (2020) ) and extended ranges to verify our method's robustness. As the experimental results in Table 11, our QHP performs stably at the standard temperature range, with $\tau = 0.05$, 0.1, and 0.2 achieving comparable mIoU scores (50.05%, 50.33%, and 49.56%, respectively). The highest mIoU (50.33%) is achieved at $\tau = 0.1$. Performance only drops noticeably when $\tau$) exceeds 0.2, aligning with observations in prior works. Collectively, these results confirm QHP's robustness to variations in the temperature parameter and eliminate the need for tedious fine-tuning within standard ranges, thereby enhancing its practicality.

| $\tau$ | 0.02 | 0.05 | 0.1 | 0.2 | 0.25 | 0.3 |
|:---:|:---:|:---:|:---:|:---:|:---:|:---:|
| mIoU (%) | 49.29 | 50.05 | **50.33** | 49.56 | 48.80 | 45.71 |

Table 11: Impact of temperature parameter $\tau$ in PC loss.

**Fairness comparison of PC Loss and standard Contrastive Loss**. To fairly compare the effectiveness of our proposed PC loss compared to standard contrastive loss, we run with the two losses under different $\lambda$ across a wide range of values from 0 to 0.9. As shown in Table 12, our PC loss consistently outperforms contrastive loss across the entire hyperparameter range, indicating the robustness and effectiveness of our PC loss.

**Trend in the Number of Bad Hubs under different Losses.** Figure 7 depicts the variation trend of bad hub points during training, with the total number of hubs fixed at 100. (1) With our PC Loss, the curve clearly shows that the number of bad hubs drops rapidly in the early training stages and later gradually stabilizes. This experimental observation is consistent with the design goal of the Prototype Distribution Optimization (PDO) module, validating its efficacy in alleviating the adverse effects caused by bad hubs. Specifically, early in training, the PDO module rectifies misclassified bad hubs via purity-reweighted contrastive loss, gradually aligning them with correct class centers. As training proceeds, the HPG module strengthens semantic association modeling, reducing bad hubs and mitigating prototype bias. In later stages, the stabilization of bad hub counts indicates that the model has captured core features of each class, thereby boosting the performance of FS-3DSeg. (2) When replacing our PC loss with standard contrastive loss, Results show that our PC Loss consistently outperforms standard contrastive loss in reducing low-purity "bad hubs" across all settings, directly validating its efficacy in mitigating hub-related misalignment—a core limitation addressed by purity-weighted correction.

| $\lambda$ | 0.9 | 0.7 | 0.5 | 0.3 | 0.1 | 0.05 | 0 |
|---|---|---|---|---|---|---|---|
| Contrastive Loss | 37.33 | 44.00 | 44.34 | 42.58 | **49.82** | 47.70 | 47.37 |
| Our PC Loss | **38.28** | **44.06** | **45.08** | **44.10** | **50.33** | **48.91** | **47.37** |

Table 12: Comparison of Contrastive Loss and Our PC Loss across different $\lambda$ values

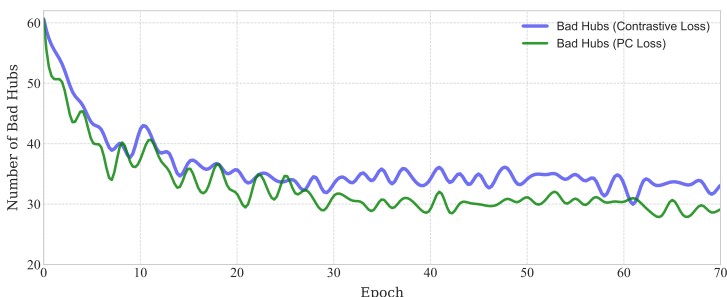

Figure 7: Trends in the number of bad hubs during training.

## A.3 COMPUTATIONAL COMPLEXITY

**Overall Computation Cost**. In Table 13, we present the number of parameters and computational complexity of our proposed method. Compared to the baseline COSeg, our approach does not introduce additional parameters but achieves higher performance. Notably, our method effectively reduces inference time and improves the FPS value. This is because the farthest point sampling (FPS) strategy adopted in COSeg requires substantial time to search for the farthest points from the support set, whereas our model can quickly sample prototype centers through the hub point mining (HPM) method, thus achieving faster speed and optimal performance.

| Methods | #Params | Inference Time (s) | FPS | FLOPs (G) | mIoU (%) |
|---|---|---|---|---|---|
| COSeg[†] | 6.11M | 0.27 | 3.66 | 14.20 | 45.93 |
| QHP | 6.11M | 0.26 | 3.83 | 14.64 | **50.33** |

Table 13: Analysis of computational cost and experimental results on the S3DIS dataset under 1-way 1-shot $S^0$ setting.

**Detailed computation cost of bipartite graph construction and clustering**. As shown in Table 14, we analyze the detailed computation cost of bipartite graph construction and clustering in 1/2/5-way settings. Experimental results confirm that our method scales efficiently, supported by three key observations: (1) Low overhead in 5-way scenarios: In 5-way 1-shot task, the computational costs of bipartite graph construction and clustering account for only 8.2% and 6.1% of the total FLOPs, respectively. The overwhelming majority of computation (over 85%) remains dominated by the backbone network. (2) Moderate scaling with complexity: Absolute computational costs grow with ways/shots, but our method scales efficiently—their relative overhead remains constrained. For example, bipartite graph FLOPs rise from 234.88 M (1-way 1-shot) to 1.583 B (5-way 1-shot), yet its overhead only increases from 1.6% to 8.2%; clustering overhead grows from 0.5% to 6.1%. Total model FLOPs merely increase from 14.64 G to 19.22 G, avoiding exponential growth. (3) Practical inference efficiency retained: For 5-way 1-shot tasks, the two components add only 12.672 s and 0.085 s of inference time, not compromising overall efficiency, suggesting that the method remains efficient for larger few-shot tasks.

## A.4 ABLATION STUDIES ON SCANNET DATASET

To analyze the hyperparameter sensitivity of QHP on ScanNet, we conduct additional ablation experiments on ScanNet under the 1-way 5-shot $S^1$ setting. We systematically analyze the effects of core hyperparameters on ScanNet, including the hub ratio, $k$ and $\eta$ in the HPG module, $k$ and $\gamma$

| Settings | Total | | Bipartite Graph | | Clustering | |
|---|---|---|---|---|---|---|
| | FLOPs (G) | FPS | FLOPs | Infer Time (s) | FLOPs | Infer Time (s) |
| 1-way 1-shot | 14.64 | 3.83 | 234.88 M (1.6%) | 4.775 | 78.64 M (0.5%) | 0.048 |
| 2-way 1-shot | 15.21 | 3.57 | 473.96 M (3.1%) | 8.189 | 157.29 M (1.0%) | 0.030 |
| 1-way 5-shot | 16.94 | 3.23 | 1.199 B (7.1%) | 5.816 | 393.22 M (2.3%) | 0.047 |
| 2-way 5-shot | 19.83 | 2.86 | 2.419 B (12.2%) | 8.169 | 786.43 M (4.0%) | 0.067 |
| 5-way 1-shot | 19.22 | 2.94 | 1.583 B (8.2%) | 12.672 | 1.1796 B (6.1%) | 0.085 |

Table 14: Detailed computation cost of bipartite graph construction and clustering under different few-shot settings on the S3DIS dataset.

in the PDO module, and the coefficient $\lambda$ for the PC loss. These experiments focus on analyzing hyperparameter consistency and cross-dataset robustness.

**Effects of Hub Prototypes from HPG**. As shown in Table 15, the optimal performance on ScanNet is achieved at a hub prototype ratio of 100%, consistent with the optimal ratio on S3DIS. Moreover, the overall trend of performance variation with respect to the hub ratio closely mirrors that observed on S3DIS, indicating similar behavior across datasets.

| Hub Ratio | 0% | 20% | 50% | 70% | 100% |
|---|---|---|---|---|---|
| mIoU (%) | 43.57 | 41.23 | 41.04 | 43.67 | **44.11** |

Table 15: Analyze the ratio of hub prototypes to total prototypes in HPG on ScanNet.

**Impact of Parameters $k$ and $\eta$ in HPG**. We conduct a joint ablation study of the HPG module on hyperparameters $k$ (columns) and $\eta$ (rows), as shown in Table 16. The results indicate that $k = 5$ achieves the best performance. Although $\eta = 150$ slightly outperforms $\eta = 100$ on ScanNet, we adopt $\eta = 100$ to ensure a fair comparison with the baseline COSeg. Notably, COSeg's prototype number ablation under the 1-way 5-shot setting on S3DIS shows a similar trend, with $\eta = 150$ outperforming $\eta = 100$ by approximately 2 points. Overall, both datasets exhibit a consistent trend with respect to variations in $k$ and $\eta$.

**Impact of Parameters $k$ and $\gamma$ in PDO**. We additionally evaluate the PDO module on hyperparameters $k$ and $\gamma$, as shown in Table 17. The optimal performance on ScanNet is achieved at $k = 5$ and $\gamma = 0.6$, consistent with S3DIS. The performance trends across $k$ and $\gamma$ are also similar between datasets, demonstrating stable behavior of the PDO module.

| mIoU (%) | 50 | 100 | 150 | 200 |
|---|---|---|---|---|
| **5** | 43.97 | 44.11 | 46.23 | 43.84 |
| **7** | 41.67 | 43.38 | 46.10 | 43.31 |
| **10** | 39.76 | 42.21 | 44.31 | 43.01 |

| mIoU (%) | 0.3 | 0.6 | 0.8 |
|---|---|---|---|
| **5** | 44.26 | 44.80 | 44.48 |
| **7** | 43.44 | 43.95 | 44.32 |
| **12** | 41.90 | 41.94 | 42.97 |

Table 16: Parameter Sensitivity of HPG (column: $k$, row: $\eta$) on ScanNet.

Table 17: Parameter Sensitivity of PDO (column: $k$, row: $\gamma$) on ScanNet.

**Analysis of the coefficient $\lambda$ for PC Loss**. We also analyze the effect of the coefficient $\lambda$ for $\mathcal{L}_{PC}$ in total loss, as shown in Table 18. The optimal performance on ScanNet is obtained at $\lambda = 0.1$, consistent with S3DIS. The overall performance trend with varying $\lambda$ also closely follows that observed on S3DIS.

In summary, the experimental results indicate that although QHP exhibits some sensitivity to hyperparameters, its generalization across datasets is robust—this is primarily supported by the remarkably consistent optimal hyperparameters across S3DIS and ScanNet, which are determined as hub ratio=100%, $k$=5, and $\gamma$=0.6, and even when deviating from these optimal values, the performance variation trends of the two datasets remain nearly identical. The modest performance differences observed on ScanNet stem from dataset-specific traits (*e.g.*, scene complexity, point cloud density variations, more diverse categories) rather than suboptimal hyperparameter configuration, meaning

| $\lambda$ | 0.7 | 0.5 | 0.3 | 0.1 | 0.05 | 0 |
|---|---|---|---|---|---|---|
| mIoU (%) | 41.19 | 42.98 | 42.77 | **44.80** | 44.35 | 44.11 |

Table 18: Impact of coefficient $\lambda$ for $L_{PC}$ on ScanNet.

QHP does not require dataset-specific hyperparameter tuning for ScanNet and its hyperparameter sensitivity does not pose a barrier to cross-dataset generalization. Consequently, the default parameter settings of QHP are broadly applicable for indoor scene datasets.

## A.5 QUALITATIVE RESULTS

**More Qualitative Results**. As shown in Figure 8, we present more results of QHP (6th column) compared to COSeg (5th column). QHP consistently demonstrates superior segmentation performance across various scenarios. For instance, in the cases of chairs and bookcases, QHP can more accurately delineate object boundaries, while COSeg tends to produce fragmented or incomplete masks. In cluttered environments, such as walls and floors, QHP is more effective at preserving the overall structure, reducing redundancy, and avoiding ambiguity near class boundaries. This ensures that QHP produces a cleaner and more precise representation compared to COSeg, especially in terms of object completeness. These results underscore QHP's substantial improvements in object boundary delineation and category shape representation. Its ability to capture finer details and reduce redundancy allows for more precise segmentation.

**Failure Cases**. In order to comprehensively demonstrate the segmentation result of the proposed method, we also give some failure cases. As shown in Figure 9, our QHP framework exhibits limitations in challenging scenarios: over-segmentation artifacts arise near complex wall structures, mask incompleteness is observed for irregular or open-form bookcases, and under-segmentation occurs on floor regions with texture transitions or interference from small objects. These failure cases are inherent to few-shot 3D point cloud segmentation tasks, particularly when faced with sparse support data and high scene complexity—limitations that are not unique to our method and do not undermine our core motivation of mitigating prototype bias. Despite these limitations, QHP still achieves superior mIoU performance, demonstrating the effectiveness of our query-aware prototype adaptation strategy in addressing the key issue of prototype bias.

## A.6 ROBUSTNESS VERIFICATION UNDER SIGNIFICANT DOMAIN SHIFT

Beyond the aforementioned FS-3DSeg experiments, we designed a cross-dataset and cross-category scenario (maximizing domain discrepancy) to evaluate our method's robustness under significant domain shift. We designed a cross-dataset and cross-category scenario (maximizing domain discrepancy) that aligns with your request for "significant domain shift". Unlike existing domain shift research in few-shot 3D segmentation (FS-3Dseg) Xiao et al. (2025), which typically focuses on dataset adaptation with overlapping classes, our cross-dataset and cross-category scenario introduces a more pronounced domain shift.

| Domain | Dataset | cvfold | Classes |
|---|---|---|---|
| Source | S3DIS | 0 | beam, board, bookcase, ceiling, chair, column |
| | | 1 | door, floor, sofa, table, wall, window |
| Target | ScanNet | 0 | bathtub, bed, bookshelf, cabinet, chair, counter, curtain, desk, sink, shower curtain |
| | | 1 | otherfurniture, picture, refrigerator, floor, door, sofa, table, toilet, wall, window |

Table 19: Partitioning and configuration of datasets in domain shift scenario.

(1) Experimental Setup: Source domain = S3DIS, target domain = ScanNet, with non-overlapping categories between the two domains to ensure a strict domain discrepancy, as shown in Table 19.

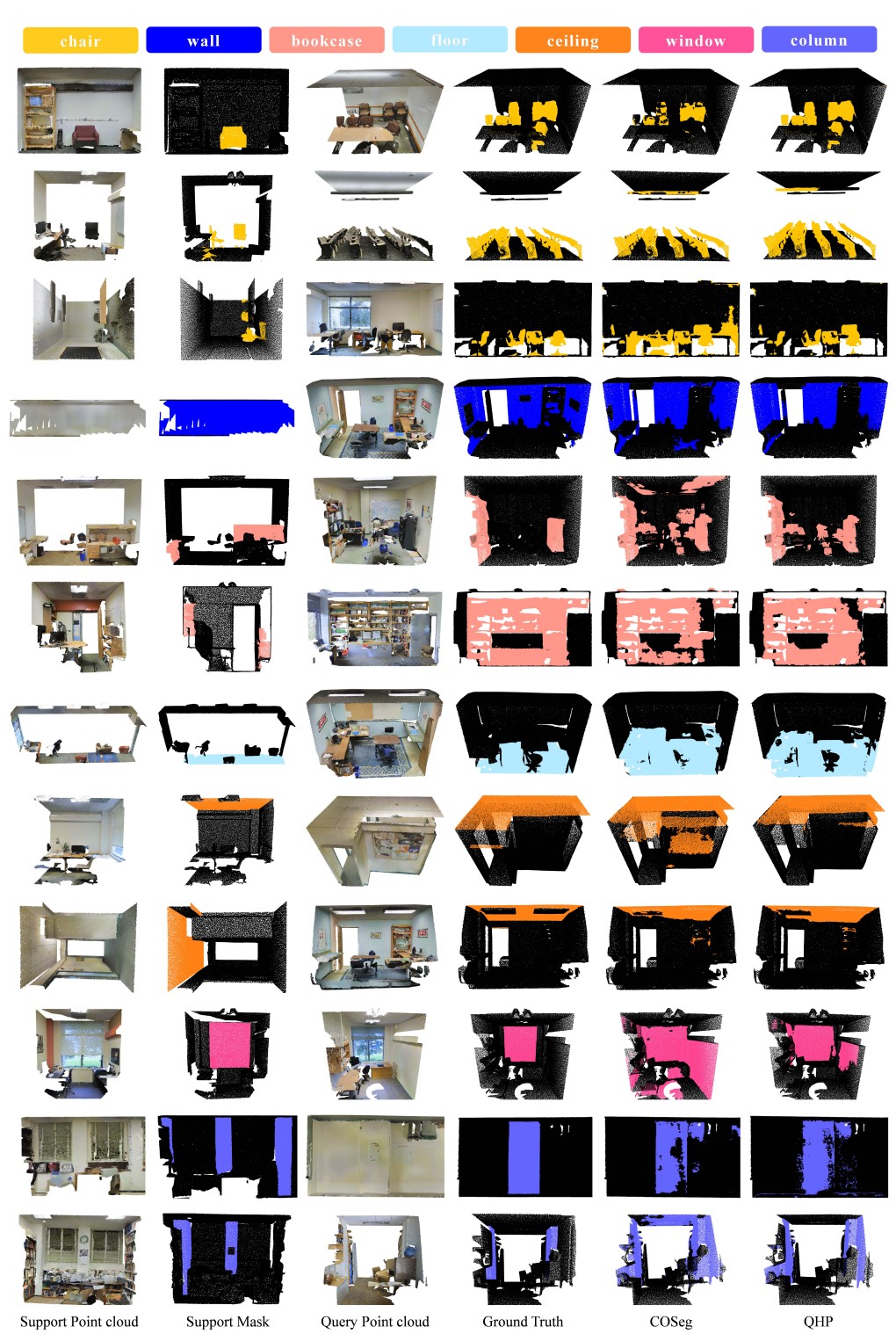

Figure 8: More qualitative comparisons between our proposed model QHP and COSeg.

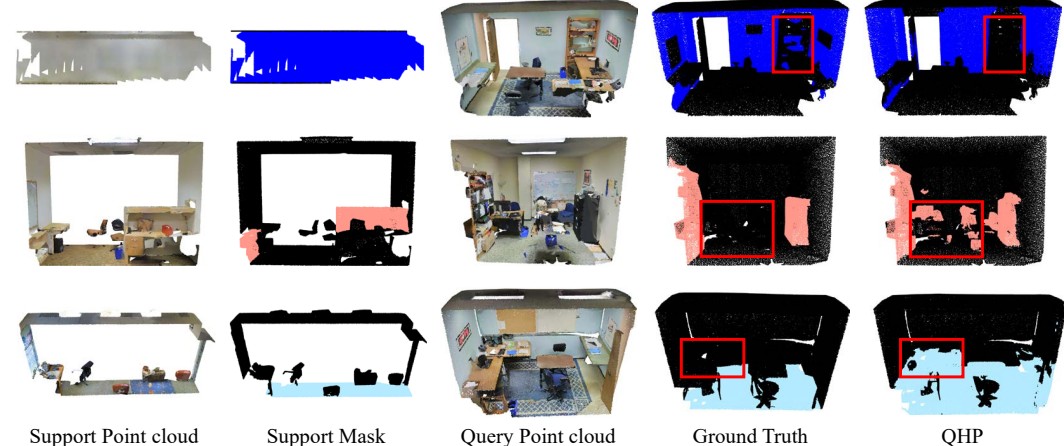

| Support Point cloud | Support Mask | Query Point cloud | Ground Truth | QHP |

Figure 9: Failure cases between our proposed model QHP and ground truth. Each row, from top to bottom, represents the 1-way 1-shot task with the target category as wall (Klein blue), bookcase (pink), and floor (pale blue), respectively.

| Methods | $S^0$ | $S^1$ | mean |
|---|---|---|---|
| Source only | 22.06 | 17.07 | 19.57 |
| **QHP** | **22.61** | **20.43** | **21.52** |
| Target | 31.58 | 29.75 | 30.67 |

Table 20: Results under domain shift scenario in terms of mIoU.

(2) Baseline & Evaluation: We adapted SegPN (Zhu et al., 2024) into a multi-prototype method (via Farthest Point Sampling, consistent with COSeg) as the baseline, evaluating under 1-way 1-shot settings.

(3) Results Analysis: As shown in Table 20, even under strict cross-domain conditions, integrating our Hub Prototype Generation (HPG) module improves mean mIoU by 1.95% over the source-only baseline. These results demonstrate that our method effectively mitigates domain bias and enhances robustness under significant domain shift.

