# OpenReview forum: "Query-aware Hub Prototype Learning for Few-Shot 3D Point Cloud Semantic Segmentation"
_ICLR.cc/2026/Conference — Submitted to ICLR 2026_

### Official Review · Reviewer_B2Tj · 2025-10-19

**Soundness:** 3
**Presentation:** 3
**Contribution:** 3
**Rating:** 8
**Confidence:** 5

**Summary:**

This paper proposes Query-aware Hub Prototype learning for few-shot 3D point cloud semantic segmentation (FS-3DSeg) to address prototype bias. The framework includes two main components: (1) a Hub Prototype Generation (HPG) module that constructs a bipartite graph between support and query points, identifies frequently linked hub points, and clusters them into query-relevant prototypes; and (2) a Prototype Distribution Optimization module that refines bad hubs with a purity-reweighted contrastive loss. Experiments on S3DIS and ScanNet show consistent improvements over other baselines.

**Strengths:**

1. The paper is clearly written and easy to follow, with strong motivation and clear figures to explain the technical details.
2. Introducing query-aware hub prototypes is a novel perspective that leverages the hubness phenomenon to address the prototype bias for FS-3DSeg.
3. The method achieves superior performance over other baseline methods, with ablation studies validating each module’s contribution.

**Weaknesses:**

1. In Hub Point Mining (line 233-235), query points are used as centers, but in Bad Hub Selection (line 254-256), query and support points are merged. The rationale for this inconsistency is unclear.
2. For the Purity-reweighted contrastive loss (Eq. 7), the weighting scheme suggests the weight for bad hubs is smaller than the fixed weight 1 for foreground prototypes while line 274-277 writes the bad hubs need stronger guidance for the alignment. The smaller weights do not align with the motivations for “stronger guidance” for bad hubs.
3. Missing related work. Recent works on 3D few-shot learning (e.g., Generalized Few-shot 3D Point Cloud Segmentation with Vision-Language Model, CVPR 2025) are highly relevant and should be cited.
4. Some grammar errors (e.g., “After that, we identify all potential bad hubs within H, which with stronger connections to center points belonging to different classes,” at line 257).

**Questions:**

Please refer to the weakness part and address the concerns there.

---

> ### Author Response · Authors · 2025-11-24
> **Response to reviewer B2Tj**
>
> Dear Reviewer B2Tj,
>
> We would like to express our gratitude for taking the time to review our manuscript and providing detailed and constructive feedback. We appreciate your positive reception of our work and carefully considered each of your points and would like to address your comments as follows:
>
> ## **W1. Rationale for the inconsistency of center points in Hub Point Mining (line 233-235) and Bad Hub Selection (Lines 254–256).**
>
> Thank you for pointing out this concern. The inconsistent center point design is a deliberate choice aligned with each module's unique objectives and the scopes of hub points to be mined, as detailed below:
>
> - In **Hub Point Mining**, we aim to select support hubs highly relevant to query points to generate prototypes for guiding query-set segmentation. **This stage focuses on query-driven hub selection and prioritizes strong prototype-query relevance**, so we only use query points as centers.
>
> - As an integral component of Prototype Distribution Optimization, **Bad Hub Selection* aims to comprehensively identify boundary-adjacent bad hubs. This enables PDO to optimize irrational distribution of bad hubs, thereby enhancing inter-class discriminability. **Since bad hubs occur both at query-support boundaries and at inter-class boundaries within the support set**, restricting centers to query points alone would miss critical support-set hubs and compromise class boundary separation. We therefore merge query and support points as centers to construct a global association graph, enabling the global selection of bad hubs.
>
> Correspondingly, we have revised the relevant sentences (Lines 239–241 and 260–261) in the revised manuscript.
>
> ## **W2. Clarification on the weighting scheme for bad hubs in the Purity-reweighted contrastive loss (Eq. 7).**
>
> We sincerely appreciate your insightful feedback and apologize for any confusion caused by our unclear description. To clarify, the weighting scheme integrated into Eq. 7 is fully aligned with our core motivation. Notably, assigning weights less than 1 to bad hubs does not "reduce" their optimization priority; instead, it enables precise correction of their semantic misalignment. The weight $w(a)$ is directly determined by each hub’s purity, and this design logic is explicitly tied to our motivation:
>
> - For high-purity hubs, which already reside in semantically reliable regions, their larger purity values map to weights closer to 0. This design ensures they undergo only subtle refinement, preserving their inherent semantic distributions.
>
> - Low-purity (bad) hubs suffer from severe semantic misalignment (e.g., class boundary ambiguity, query irrelevance). Their smaller purity values result in weights closer to 1, which amplifies the contrastive loss applied to these hubs—providing stronger alignment supervision to correct their deviations.
>
> ## **W3. Missing related work.**
>
> We sincerely appreciate your valuable suggestion. Following your guidance, we have incorporated the recommended relevant studies into Section 2 ("Related Work", Line 138-139) of the revised manuscript, with details below:
>
> "Besides, recent generalized FS-3DSeg methods [1] leverage vision-language models."
>
> ## **W4. Some grammar errors.**
>
> Thank you for your helpful feedback. We sincerely apologize for the oversights in grammatical expression. We have conducted a thorough revision of all identified grammar errors in the revised manuscript. Specifically, the problematic sentence has been revised in (Lines 264-265) in the manuscript, detailed as below:
>
> "After that, we identify all potential bad hubs within $H$ that have stronger connections to center points belonging to different classes."
>
> References:
>
> [1] Zhaochong An, Guolei Sun, Yun Liu, Runjia Li, Junlin Han, Ender Konukoglu, and Serge Belongie. Generalized few-shot 3d point cloud segmentation with vision-language model. In Proceedings of the IEEE/CVF Conference on Computer Vision and Pattern Recognition (CVPR), pp. 16997–17007, 2025a.

---

### Official Review · Reviewer_63hN · 2025-10-31

**Soundness:** 2
**Presentation:** 3
**Contribution:** 3
**Rating:** 4
**Confidence:** 4

**Summary:**

The paper proposes a Query-aware Hub Prototype (QHP) approach for few-shot 3D point-cloud segmentation. It introduces (i) Hub Prototype Generation (HPG), selecting “hub” support points via neighborhood popularity, then performing local clustering to form multiple prototypes per class while interacting with query features; and (ii) Prototype Distribution Optimization (PDO), identifying “bad hubs” by purity and applying a purity-weighted contrastive loss to pull them toward class centers.

**Strengths:**

The motivation for addressing prototype bias from using only support features is interesting.

The experimental comparison is sufficient.

Stepwise ablations (Baseline → +HPG → +PDO) indicate incremental improvements

**Weaknesses:**

1. The abstract and introduction state that “existing metric-based prototype learning methods generate prototypes solely from the support set, without considering their relevance to query data.” This is not generally accurate. A substantial body of transductive / query-guided work explicitly leverages unlabeled query features to refine or align prototypes with the query distribution, e.g., support–query alignment and query-aware refinement [1, 2, 3]. Because this statement underpins the paper’s motivation and novelty positioning, it weakens the contribution as written.

2. The claimed benefits of hub selection and purity-weighted correction are motivated empirically. There is no analysis of hub statistics, misalignment probability, or even simple bounds showing when purity-weighted attraction is preferable to standard contrastive objectives.

3. Why do you prioritize pulling low-purity (“bad-hub”) / outlier prototypes toward the class center instead of the complementary strategy of reinforcing high-purity (“good-hub”) prototypes toward positives? What theoretical or empirical evidence shows that focusing on repairing bad hubs yields better decision boundaries than amplifying good hubs?

4. The improvements of most settings in Tables 1 and 2 are quite marginal. In Table 6, the authors perform a comparison between PC loss and contrastive loss. Do you use the optimal hyperparameter (e.g., $\lambda$) for the standard contrastive loss? The performance difference between PC loss and contrastive loss is marginal, and your Fig. 5 (c) shows that different $\lambda$ have a critical influence on the performance. Will using a better hyperparameter for contrastive loss lead to better performance?

5. Figure 5 shows that the performance of the proposed method is highly sensitive to the hyperparameters, $k$, $\eta$, $\lambda$, and $\gamma$. It’s unclear how robust the method is for a new dataset without re-tuning.

6. What is the performance when using more prototypes than 100?



[1]. Wang K, Liew J H, Zou Y, et al. Panet: Few-shot image semantic segmentation with prototype alignment[C]//proceedings of the IEEE/CVF international conference on computer vision. 2019: 9197-9206.

[2]. D. Hu, S. Chen, H. Yang and G. Wang, "Query-Guided Support Prototypes for Few-Shot 3D Indoor Segmentation," in IEEE Transactions on Circuits and Systems for Video Technology, vol. 34, no. 6, pp. 4202-4213, June 2024.

[3]. Ning Z, Tian Z, Lu G, et al. Boosting few-shot 3d point cloud segmentation via query-guided enhancement[C]//Proceedings of the 31st ACM international conference on multimedia. 2023: 1895-1904.

**Questions:**

see my weakness.

---

> ### Author Response · Authors · 2025-11-24
> **Response to reviewer 63hN (1/2)**
>
> Dear Reviewer 63hN,
>
> Thank you for your comments and for recognizing our interesting motivation, stepwise ablations, and sufficient experimental comparison. Below, we address your mentioned weaknesses (W) and questions (Q), referring to modifications made in the manuscript (highlighted in blue). Finally, we provide a brief summary.
>
> ## **W1: Inaccurate Claim About Existing Query-Guided Methods.**
>
> Thank you for this comment and the opportunity to clarify our claim and novelty.
>
> While some methods [1,2,3] do leverage query features, there exists a fundamental architectural difference between existing approaches and ours. Common transductive/query-guided methods, such as support–query alignment and query-aware refinement, follow a two-stage paradigm: they first generate prototypes solely from support data, then refine or align these prototypes using query information in a subsequent stage. However, as the initial prototypes have already deviated significantly from the query, making subsequent corrections challenging.
> In contrast, our approach integrates query-awareness \textbf{directly} into the prototype generation process itself, enabling true end-to-end optimization without intermediate refinement steps. We have expanded our motivation discussion (in lines 15-18 and Lines 52-72 in the revised manuscript) to more accurately describe our approach and the differences between these methods.
>
> ## **W2: Empirical Evidence for Purity-Weighted Correction.**
>
> We sincerely appreciate your insightful suggestion. To validate the benefits of our PC Loss over standard Contrastive Loss, we supplement targeted experiments and analyses:
>
> **Prototype Distributions and Class Bounds:** We add t-SNE visualizations to compare prototype distributions under two losses, as shown in Figure 6 (c)–(d) in the revised manuscript. Results demonstrate that while standard contrastive loss marginally improves separability and mitigates ambiguity, residual confusion persists near decision boundaries. In contrast, our PC Loss yields more compact intra-class clusters and sharper inter-class boundaries via explicit prototype alignment.
>
> **Bad Hub Statistics**:
> We present quantitative comparisons of bad hub counts under the two losses, as shown in Figure 7 in the revised manuscript. Results show that our PC Loss consistently outperforms standard contrastive loss in reducing low-purity "bad hubs" across all settings, directly validating its efficacy in mitigating hub-related misalignment—a core limitation addressed by purity-weighted correction.
>
> These complementary visual and quantitative results provide compelling empirical support for the practical advantages of PC Loss, reinforcing the rationale behind our purity-weighted correction strategy. Detailed analyses are provided in the revised manuscript.
>
>
> ## **W3: Why Pull Bad Hubs toward the Class Center Rather Than Good Hubs?**
>
> We thank your insightful question. Our design choice is motivated by both theoretical considerations and empirical evidence:
>
> **Theoretical Analysis:** Intuitively, good-hub prototypes (high purity) already exhibit strong intra-class compactness and optimal alignment with class centers. Further reinforcing them toward positives yields diminishing returns. Moreover, excessive reinforcement may collapse the beneficial intra-class variance crucial for maintaining inter-class discrimination. Conversely, bad-hub or outlier prototypes are the main causes of boundary ambiguity, as they often lie in problematic areas between classes where errors occur.
>
> **Experimental Evidence:** We added comparative experiments (our "pulling bad hubs" vs. alternative "pulling good hubs"), with details provided in Table 10 and Section A.2 (Impact of Different Anchors in PC Loss) of the revised manuscript. Quantitatively, our strategy achieves 50.33\% mIoU, outperforming "pulling good hubs" by 2.5\% mIoU (47.83\%). This confirms that prioritizing bad-hub correction yields better decision boundaries than reinforcing good hubs, validating the superiority of our design.

---

> ### Author Response · Authors · 2025-11-24
> **Response to reviewer 63hN (2/2)**
>
> ## **W4: Performance and Hyperparameter Tuning Fairness.**
>
> **On the improvements:** Our method achieves notable performance gains on S3DIS, with particularly prominent improvements in the 1-way few-shot setting. These gains are particularly significant for few-shot 3D segmentation. On the more challenging ScanNet dataset, characterized by complex point cloud structures, numerous object categories, and significant scene variability, our approach still achieves consistent improvements across all settings, highlighting its superior robustness and adaptability to diverse 3D environments.
>
> **On the hyperparameter fairness:** We appreciate your concern about fair comparison. To address this point, we conducted a comprehensive hyperparameter sweep for $\lambda$ across a wide range of values from 0 to 0.9 for both contrastive loss and our proposed PC loss (updated in Table 12 in Appendix A.2). Below, we show the detailed table results for your convenience. The results show that our PC loss demonstrates consistent superiority in robustness over contrastive loss across the entire hyperparameter range. Note that both losses share a similar trend as $\lambda$ increases, while obtaining the best optimal results when $\lambda=0.1$. These results validate that the benefit stems from the design of our PC loss by purity-reweighted scheme, instead of hyperparameter tuning.
>
>
> | $\lambda$           | 0.9   | 0.7   | 0.5   | 0.3   | 0.1   | 0.05  | 0    |
> |------------|-------|-------|-------|-------|-------|-------|------|
> |  Contrastive Loss  | 37.33 | 44.00 | 44.34 | 42.58 | **49.82** | 47.70     | 47.37 |
> | Our PC Loss | **38.28** | **44.06** | **45.08** | **44.10** | **50.33** | **48.91** | **47.37** |
>
> ## **W5: Hyperparameter Robustness.**
>
> We appreciate the robustness concern. In Figure 5, we explored hyperparameter sensitivity on S3DIS. To address this concern, we have conducted comprehensive experiments on ScanNet to analyze hyperparameter consistency and cross-dataset robustness (Section A.4 in the revised manuscript). These ScanNet experiments confirm that optimal hyperparameters remain remarkably consistent across datasets (hub ratio=100\%, $k$=5, $\gamma$=0.6). Beyond optimal values, the performance trends are nearly identical between datasets, thereby proving the robustness of our method for deployment on new datasets. Below we show the tables for your convenience.
>
> Table 15 in Section A.4. Analyze the ratio of hub prototypes
> to total prototypes.
> | Hub Ratio | 0\%   | 20\%  | 50\%  | 70\%  | 100\%   |
> |-----------|-------|-------|-------|-------|---------|
> | mIoU (\%) | 43.57 | 41.23 | 41.04 | 43.67 | **44.11** |
>
> Table 16 in Section A.4. Parameter Sensitivity of HPG (column: $k$,row:$\eta$)
> | mIoU (\%) | 50    | 100   | 150   | 200   |
> |------------|-------|-------|-------|-------|
> | 5          | 43.97 | 44.11 | **46.23** | 43.84 |
> | 7          | 41.67 | 43.38 | 46.10 | 43.31 |
> | 10         | 39.76 | 42.21 | 44.31 | 43.01 |
>
> Table 17 in Section A.4. Parameter Sensitivity of PDO
> (column: $k$, row: $\gamma$).
> | mIoU (\%) | 0.3   | 0.6   | 0.8   |
> |--------------|-------|-------|-------|
> | 5        | 44.26 | **44.80** | 44.48 |
> | 7        | 43.44 | 43.95 | 44.32 |
> | 12       | 41.90 | 41.94 | 42.97 |
>
> Table 18 in Section A.4. Analysis of the temperature parameter $\lambda$ for $\mathcal{L}_\text{PC}$
> | $\lambda$    | 0.7 | 0.5 | 0.3 | 0.1 | 0.05 | 0   |
> |------------|---------|---------|---------|---------|----------|---------|
> | mIoU (\%)   | 41.19   | 42.98   | 42.77   | **44.80**   | 44.35    | 44.11   |
>
>
> ## **W6: Performance on more prototypes than 100?**
>
> Thank you for raising this concern. We actually have provided results for using more than 100 prototypes in our submission, as shown in Figure 5(a) and Table 7 in revised Appendix A.2. We acknowledge that the column headers were incorrectly labeled as "0.3 / 0.45 / 0.6 / 0.8" when they should correspond to approximately "20 / 50 / 100 / 150" prototypes, respectively. We have corrected this labeling error in the revision, though the experimental results remain accurate.
>
> In Figure 5(a) and Table 7, the results clearly show that performance does not improve beyond 100 prototypes. For example, under \(k = 5\), using around 100 prototypes achieves 47.37\% mIoU, whereas increasing to 150 prototypes reduces performance to 43.24\%, indicating that overly large prototype sets introduce redundancy and degrade effectiveness. Besides, updated ablation experiments for ScanNet's 1-way 5-shot $S^1$ ablation experiments (Table 16 in revised Appendix A.4) demonstrate that increasing the number of prototypes to 150 enhances performance.

---

### Official Review · Reviewer_F9pR · 2025-10-31

**Soundness:** 2
**Presentation:** 2
**Contribution:** 2
**Rating:** 4
**Confidence:** 4

**Summary:**

This paper introduces a novel method for few-shot 3D point cloud semantic segmentation (FS-3DSeg) by proposing a Query-aware Hub Prototype (QHP) learning framework. The core idea is to mitigate prototype bias in metric-based few-shot methods by selecting query-relevant hub points from the support set. Two main modules are proposed: (1) Hub Prototype Generation (HPG), which constructs a bipartite graph between support and query sets to identify frequently-linked support “hub” points and generate prototypes via local clustering; and (2) Prototype Distribution Optimization (PDO), which uses a purity-reweighted contrastive loss to refine bad prototypes and enhance class compactness. Extensive experiments on S3DIS and ScanNet show improved performance over several baselines, supported by ablation studies and qualitative comparisons.

**Strengths:**

- The introduction of “hubness” for prototype generation in FS-3DSeg is novel and provides a new perspective on addressing support-query misalignment.
- Extensive experiments: Evaluations on S3DIS and ScanNet across 1-shot and 5-shot settings are comprehensive. Ablation studies and parameter sensitivity analyses support the effectiveness of the proposed modules.
- Clear writing and figures: The paper is well written, and figures (e.g., Figure 1, Figure 2, Figure 3) effectively illustrate the key concepts.

**Weaknesses:**

- Limited comparisons to recent or stronger baselines: The paper lacks comparisons with more recent state-of-the-art methods beyond COSeg and QGE/QGPA. Recent approaches that incorporate transformer-based meta learners, distillation, or prompt-based adaptation are not discussed or evaluated.
- Unclear generalization and robustness of hub mining: The “hubness” concept is somewhat heuristic and heavily relies on k-NN and purity thresholds. There is limited discussion or analysis on whether hub mining is robust under significant domain shift, class imbalance, or in real-world open-set scenarios.
- Efficiency claims are underexplored: While FLOPs and inference time are briefly compared, the computational cost of bipartite graph construction and clustering in large-scale 5-way settings is not analyzed. It is unclear whether the method truly scales well.
- No failure cases shown: All qualitative visualizations emphasize improvement. It would be helpful to include cases where the method fails or introduces errors to better understand its limitations.
- Motivation example could be more intuitive: Although prototype bias is well described in text, there is no concrete visual example showing how traditional support-only prototypes fail on specific query samples.

**Questions:**

See weaknesses

---

> ### Author Response · Authors · 2025-11-24
> **Response to reviewer F9pR (1/3)**
>
> Dear Reviewer F9pR,
>
> We would like to express our gratitude for taking the time to review our manuscript and providing detailed and constructive feedback. We carefully considered each of your points and would like to address your comments as follows:
>
> ## **W1. Limited comparisons.**
>
> Thank you for this valuable comment. We appreciate the opportunity to clarify and strengthen our baseline comparisons:
>
> (1) Comparisons with recent FS-3DSeg methods: Our original submission already includes SOTA baselines (COSeg, QGE, QGPA, SegPN). Recently, there have been some multi-modal FS-3DSeg methods (e.g., RCHP [1], MM-FSS [2]). Direct comparison with these works is unfair, as our work adheres to a stricter point-only setting while they use additional 2D/text supervision. Nevertheless, preliminary comparisons show our QHP is competitive with MM-FSS [2] in many settings and outperforms it in 1-way 1-shot/5-shot $S^0$ settings.
>
> (2) Discussion on suggested methods: Following your suggestion, we reviewed transformer-based ([3], [4]), distillation-based ([5], [6]), and prompt-driven ([7], [8]) works. These are not directly comparable to our FS-3DSeg setting: some require additional modalities (text/language [7], [8]), target distinct domains (2D images [3], outdoor LiDAR [4], medical imaging [5]), and [6] is closed-source (precluding fair reproduction). We acknowledge their insights and agree that adapting such techniques to FS-3DSeg is a promising future direction.
>
> ## **W2. Unclear generalization and robustness of hub mining.**
>
> We sincerely appreciate your insightful feedback. Below, we clarify the rationale for "hubness" and address robustness across challenging scenarios via supplementary experiments and analysis:
>
> (1) Rationale of "hubness" with core motivation:
> While "Hubness" is inherently a heuristic concept, it is not arbitrary but tailored to address the key limitation of previous works: traditional support-only prototypes are biased (outlier-sensitive, query-misaligned), stemming from their neglect of query-support cross-set relationships. In our work, "hubness" identifies support points frequently referenced by queries, enabling query-aware hub prototype generation. Specifically, k-NN captures local geometric/semantic consistency to select query-relevant hubs (mitigating "support-only bias"), while the purity threshold filters noisy bad hubs and optimizes class separability.
>
> (2) Robust under **domain shift**:
> To address domain shift concerns, we supplement experiments and analysis in the revised Appendix A.6. We designed a cross-dataset, cross-category scenario (maximizing domain discrepancy) to simulate "significant domain shift". For experimental details, please refer to Appendix A.6. Results show that our QHP improves mean mIoU by 1.95\% over the source-only baseline, even under strict cross-domain conditions. This validates our method’s ability to mitigate domain bias and enhance robustness against significant domain shift.
>
> *Table 20 in Section A.6. Adaptation results from S3DIS to ScanNet in terms of mIoU.*
> | Methods     | $S^0$  | $S^1$  | mean  |
> |-------------|--------|--------|-------|
> | Source only | 22.06  | 17.07  | 19.57 |
> | QHP         | 22.61  | 20.43  | 21.52 |
> | Target      | 31.58  | 29.75  | 30.67 |
>
> (3) Robust under **Class imbalance**:
> We politely clarify that our evaluated datasets are inherently class-imbalanced. For instance, as the table below, S3DIS exhibits a highly skewed category distribution: frequent classes appear in nearly all scans, while rare classes are severely underrepresented. Additionally, each training/test episode exhibits significant imbalance between foreground classes and between foreground/background classes, with substantial point count disparities. Despite this natural imbalance, QHP consistently improves performance across all splits, validating that our method is inherently robust to substantial class-frequency variation.
>
> *Table: Number of point cloud scans for each class in S3DIS dataset (total scans: 271).*
> | Class   | ceiling | floor | wall | beam | column | window |
> |-------------|-------------|-----------|----------|----------|------------|------------|
> | Number      | 271         | 270       | 270      | 36       | 24         | 81         |
> | **Class**   | **door**    | **table** | **chair** | **sofa** | **bookcase** | **board** |
> | Number      | 92          | 102       | 32       | 7        | 124        | 14         |
>
> (4) Robust under **Real-world open-set scenarios**:
>
> We agree that real-world open-set FS-3DSeg is a valuable direction, and we will view this as an important extension direction for our future work.
> In the submitted manuscript, since our baselines ([10], [11]) focus solely on indoor closed-set settings with no established open-set evaluation protocol, our experiments thus adhere to this setting for fair comparison.

---

> ### Author Response · Authors · 2025-11-24
> **Response to reviewer F9pR (2/3)**
>
> ## **W3. Efficiency claims are underexplored.**
>
> We sincerely appreciate your critical comment. To address your concern, we have added comprehensive experiments to analyze the detailed computation cost of bipartite graph construction and clustering, as shown in Table 14 in the revised Appendix).
>
> *Table 14 in Section A.3. Detailed computation cost of bipartite graph construction and clustering.*
> | Settings      |   Total          |        |   Bipartite Graph     |                  | Clustering   |       |
> |---|---|---|---|---|---|---|
> |               | FLOPs (G)        | FPS    | FLOPs                   | Infer Time (s)   | FLOPs             | Infer Time (s)   |
> | 1-way 1-shot  | 14.64            | 3.83 | 234.88 M (1.6\%)         | 4.775            | 78.64 M (0.5\%)     | 0.048                 |
> | 2-way 1-shot  | 15.21            | 3.57 | 473.96 M (3.1\%)         | 8.189            | 157.29 M (1.0\%)    | 0.030                 |
> | 1-way 5-shot  | 16.94            | 3.23 | 1.199 B (7.1\%)          | 5.816            | 393.22 M (2.3\%)    | 0.047                 |
> | 2-way 5-shot  | 19.83            | 2.86 | 2.419 B (12.2\%)         | 8.169            | 786.43 M (4.0\%)    | 0.067                 |
> | 5-way 1-shot  | 19.22            | 2.94 | 1.583 B (8.2\%)          | 12.672           | 1.1796 B (6.1\%)    | 0.085                 |
>
> Experimental results confirm that our method scales efficiently, supported by three key observations:
>
> - Low overhead in 5-way scenarios: In 5-way 1-shot task, the computational costs of bipartite graph construction and clustering account for only 8.2\% and 6.1\% of the total FLOPs, respectively. The overwhelming majority of computation (over 85\%) remains dominated by the backbone network.
>
> - Moderate scaling with complexity: Absolute computational costs grow with ways/shots, but our method scales efficiently—their relative overhead remains constrained. For example, bipartite graph FLOPs rise from 234.88 M (1-way 1-shot) to 1.583 B (5-way 1-shot), yet its overhead only increases from 1.6\% to 8.2\%; clustering overhead grows from 0.5\% to 6.1\%. Total model FLOPs merely increase from 14.64 G to 19.22 G, avoiding exponential growth.
>
> - Practical inference efficiency retained: For 5-way 1-shot tasks, the two components add only 12.672 s and 0.085 s of inference time, not compromising overall efficiency, suggesting that the method remains efficient for larger few-shot tasks.
>
> ## **W4. No failure cases shown.**
>
> We sincerely appreciate your suggestion. We add some failure cases in Figure 9 (presented in Section A.5 of the revised manuscript). Results show that our QHP framework exhibits limitations in challenging scenarios: over-segmentation artifacts arise near complex wall structures, mask incompleteness is observed for irregular or open-form bookcases, and under-segmentation occurs on floor regions with texture transitions or interference from small objects.
> These failure cases are inherent to few-shot 3D point cloud segmentation tasks, particularly when faced with sparse support data and high scene complexity—limitations that are not unique to our method and do not undermine our core motivation of mitigating prototype bias. Despite these limitations, QHP still achieves superior mIoU performance, demonstrating the effectiveness of our query-aware prototype adaptation strategy in addressing the key issue of prototype bias.
>
>
> ## **W5. Motivation example could be more intuitive.**
>
> We sincerely appreciate your valuable suggestion.  To strengthen the intuitiveness of our motivation, we have added t-SNE visualizations of the feature space for multiple test episodes, as shown in Figure 6 (the figure and explanation have be included in Section 4.3 of the revised manuscript). These visualizations include foreground/background features from the query set, as well as foreground/background prototypes, enabling direct comparison of prototype bias with query distributions.
> This visualization provides a concrete and intuitive example of why support-only prototypes fail and how QHP progressively mitigates prototype bias through HPG and PDO.
>
>
> - For COSeg, support-only prototypes often deviate from the true query distribution—showing both class-center shifts and boundary confusion.
>
> - After introducing our Hub Prototype Generation (HPG) module, class-center misalignment is noticeably reduced, but ambiguous regions near decision boundaries remain.
>
> - To further investigate, while retaining the HPG module, we replace our PC loss with standard contrastive loss in PDO module. Results indicate that although separability is slightly improved and boundary ambiguity is partially alleviated, residual confusion near decision boundaries remains unresolved.
>
> - With our full QHP framework, including both HPG module and the PDO module equiped with PC loss, these boundary issues are further corrected, and prototypes become more compact and better aligned with their corresponding query clusters.

---

> ### Author Response · Authors · 2025-11-24
> **Response to reviewer F9pR (3/3)**
>
> References:
>
> [1] Lili Wei, Congyan Lang, Zheming Xu, Liqian Liang, and Jun Liu. Few-shot 3d point cloud segmentation via relation consistency-guided heterogeneous prototypes. IEEE Trans. Multim., 27: 3158–3170, 2025.
>
> [2] Zhaochong An, Guolei Sun, Yun Liu, Runjia Li, Min Wu, Ming-Ming Cheng, Ender Konukoglu, and Serge J. Belongie. Multimodality helps few-shot 3d point cloud semantic segmentation. In The Thirteenth International Conference on Learning Representations, ICLR 2025, Singapore, April 24-28, 2025. OpenReview.net, 2025c.
>
> [3] Dustin Carri´on-Ojeda, Stefan Roth, and Simone Schaub-Meyer. Efficient masked attention transformer for few-shot classification and segmentation. CoRR, abs/2507.23642, 2025. doi: 10.48550/ARXIV.2507.23642. URL https://doi.org/10.48550/arXiv.2507.23642.
>
> [4] Ke Song, Yunhe Wu, Chunchit Siu, and Huiyuan Xiong. Graphgsocc: Semantic-geometric graph transformer with dynamic-static decoupling for 3d gaussian splatting-based occupancy prediction. CoRR, abs/2506.14825, 2025. doi: 10.48550/ARXIV.2506.14825. URL https://doi.org/10.48550/arXiv.2506.14825.
>
> [5] Yuqun Yang, Jichen Xu, Mengyuan Xu, Xu Tang, Bo Wang, Kechen Shu, and Zheng You. Fsvs-net: A few-shot semi-supervised vessel segmentation network for multiple organs based on feature distillation and bidirectional weighted fusion. Inf. Fusion, 123:103281, 2025. doi: 10.1016/J.INFFUS.2025.103281. URL https://doi.org/10.1016/j.inffus.2025.103281.
>
> [6] Qianguang Zhao, Dongli Wang, Yan Zhou, Jianxun Li, and Richard Irampaye. Few to big: Prototype expansion network via diffusion learner for point cloud few-shot semantic segmentation. CoRR, abs/2509.12878, 2025. doi: 10.48550/ARXIV.2509.12878. URL https://doi.org/10.48550/arXiv.2509.12878.
>
> [7] Jiahui Wang, Haiyue Zhu, Haoren Guo, Abdullah Al Mamun, Cheng Xiang, and Tong Heng Lee. Epsegfz: Efficient point cloud semantic segmentation for few- and zero-shot scenarios with language guidance, 2025b. URL https://arxiv.org/abs/2511.11700.
>
> [8] Hongyu Sun, Yongcai Wang, Wang Chen, Haoran Deng, and Deying Li. Parameter-efficient prompt learning for 3d point cloud understanding. In IEEE International Conference on Robotics and Automation, ICRA 2024, Yokohama, Japan, May 13-17, 2024, pp. 9478–9486. IEEE, 2024. doi: 10.1109/ICRA57147.2024.10610093. URL https://doi.org/10.1109/ICRA57147.2024.10610093.
>
> [9] Jiwei Xiao, Ruiping Wang, Chen He, and Xilin Chen. Cross-domain few-shot 3d point cloud semantic segmentation. Pattern Recognit. Lett., 197:51–57, 2025.
>
> [10] Jie Liu, Wenzhe Yin, Haochen Wang, Yunlu CHen, Jan-Jakob Sonke, and Efstratios Gavves. Dynamic prototype adaptation with distillation for few-shot point cloud segmentation. In International Conference on 3D Vision (3DV), pp. 810–819, 2024.
>
> [11] Changshuo Wang, Shuting He, Xiang Fang, Meiqing Wu, Siew-Kei Lam, and Prayag Tiwari. Taylor series-inspired local structure fitting network for few-shot point cloud semantic segmentation. In Toby Walsh, Julie Shah, and Zico Kolter (eds.), AAAI-25, Sponsored by the Association for the Advancement of Artificial Intelligence, February 25 - March 4, 2025, Philadelphia, PA, USA, pp. 7527–7535. AAAI Press, 2025a.

---

### Official Review · Reviewer_SJ47 · 2025-11-01

**Soundness:** 3
**Presentation:** 3
**Contribution:** 3
**Rating:** 4
**Confidence:** 4

**Summary:**

This paper addresses prototype bias in few-shot 3D point cloud semantic segmentation by introducing a Query-aware Hub Prototype (QHP) learning framework. QHP constructs class prototypes using high-frequency hub points that exhibit the highest similarity to the query point clouds. To further enhance prototype quality, the proposed Prototype Distribution Optimization (PDO) module identifies potential bad hubs, and then adopt a Purity-reweighted Contrastive (PC) loss to suppress these bad hubs and optimize the prototype distribution.

**Strengths:**

1. It is interesting to introduce the concept of hubs into few-shot 3D point cloud semantic segmentation. By selecting the parts of the support point cloud closest to the query to generate prototypes, the resulting prototypes better align with the distribution of the query point cloud.

2. The experimental evaluation is sufficiently comprehensive and convincingly supports the effectiveness of the proposed QHP.

**Weaknesses:**

1. The experiment in Table 5 suggests that QHP is relatively sensitive to hyperparameter choices, raising concerns about its generalization across different datasets. This sensitivity may also explain why the performance gains on ScanNet are less pronounced than those on S3DIS. To address this issue, please verify whether substantially different hyperparameters are required for ScanNet.

2. The temperature parameter in Equation 8 has not been subjected to ablation analysis, despite being a critical factor influencing the effectiveness of contrastive learning.

**Questions:**

I have concerns regarding the weakness, particularly the generalization of QHP. If these concerns are adequately addressed, I would be happy to raise my score.

---

> ### Author Response · Authors · 2025-11-24
> **Response to reviewer SJ47**
>
> Dear Reviewer SJ47,
>
> We would like to express our gratitude for taking the time to review our manuscript and providing detailed and constructive feedback. We carefully considered each of your points and would like to address your comments as follows:
>
>
> ## **W1. Analysis of hyperparameters for ScanNet.**
>
> Thank you for your valuable comments. To address your concern, we have conducted more ablation experiments on ScanNet under the 1-way 5-shot $S^1$ setting.
> These experiments focus on analyzing hyperparameter consistency and cross-dataset robustness, **with detailed results and analysis presented in Section A.4 (Ablation Studies on ScanNet Dataset) of the revised manuscript**.
> Specifically, **we systematically analyze the effects of core hyperparameters on ScanNet, including the hub ratio, $k$ and $\eta$ in the HPG module, $k$ and $\gamma$ in the PDO module, and the coefficient $\lambda$ for the PC loss.**
>
> The experimental results confirm that optimal hyperparameters remain remarkably consistent across S3DIS and ScanNet: hub ratio=100\%, $k$=5, $\gamma$=0.6. Even when deviating from these optimal values, the performance variation trends are nearly identical between the two datasets. This demonstrates that QHP does not require dataset-specific hyperparameter tuning for ScanNet, and its hyperparameter sensitivity is not a barrier to cross-dataset generalization. Some limited performance improvement on ScanNet stems from dataset-specific traits (e.g., scene complexity, point cloud density variations, and more diverse categories) rather than suboptimal hyperparameter configuration.
>
> For your convenience, we present the key experimental results in the table below:
>
> *Table 15 in Section A.4. Analyze the ratio of hub prototypes to total prototypes in HPG on ScanNet.*
> | Hub Ratio | 0\%   | 20\%  | 50\%  | 70\%  | 100\%   |
> |-----------|-------|-------|-------|-------|---------|
> | mIoU (\%) | 43.57 | 41.23 | 41.04 | 43.67 | **44.11** |
>
>
> *Table 16 in Section A.4. Parameter Sensitivity of HPG (column: $k$,row:$\eta$)  on ScanNet.*
> | mIoU (\%) | 50    | 100   | 150   | 200   |
> |------------|-------|-------|-------|-------|
> | 5          | 43.97 | 44.11 | **46.23** | 43.84 |
> | 7          | 41.67 | 43.38 | 46.10 | 43.31 |
> | 10         | 39.76 | 42.21 | 44.31 | 43.01 |
>
> *Table 17 in Section A.4. Parameter Sensitivity of PDO (column: $k$, row: $\gamma$) on ScanNet.*
> | mIoU (\%) | 0.3   | 0.6   | 0.8   |
> |--------------|-------|-------|-------|
> | 5        | 44.26 | **44.80** | 44.48 |
> | 7        | 43.44 | 43.95 | 44.32 |
> | 12       | 41.90 | 41.94 | 42.97 |
>
>
> *Table 18 in Section A.4. Analysis of the coefficient $\lambda$ for $\mathcal{L}_\text{PC}$ on ScanNet.*
> | $\lambda$ | 0.7 | 0.5 | 0.3 | 0.1 | 0.05 | 0   |
> |------------|---------|---------|---------|---------|----------|---------|
> | mIoU (\%)   | 41.19   | 42.98   | 42.77   | **44.80**   | 44.35    | 44.11   |
>
>
> ## **W2. Ablation analysis of the temperature parameter in Equation 8.**
>
> We sincerely appreciate your helpful suggestion and apologize for overlooking the ablation analysis of this critical parameter. To address this, we have **add an ablation study on the temperature parameter ($\tau$) in Equation 8**, covering both typical settings commonly used in contrastive learning (e.g., $\tau$ = 0.02 in [1], $\tau$ = 0.1 in [2]) and extended ranges to verify our method’s robustness.
> **We have incorporated the experimental results and key conclusions into Section A.2 (More Ablation Studies on S3DIS Dataset) of the revised manuscript.**
>
> *Table 11 in Section A.2. Impact of temperature parameter $\tau$ in PC loss.*
> | $\tau$   | 0.02  | 0.05  | 0.1    | 0.2   | 0.25  | 0.3   |
> |----------|-------|-------|--------|-------|-------|-------|
> | mIoU (\%) | 49.29 | 50.05 | **50.33** | 49.56 | 48.80 | 45.71 |
>
> The experimental results show that QHP performs stably at the standard temperature range, with $\tau$ = 0.05, 0.1, and 0.2 achieving comparable mIoU scores (50.05\%, 50.33\%, and 49.56\%, respectively). The highest mIoU (50.33\%) is achieved at $\tau$ = 0.1. Performance only drops noticeably when $\tau$ exceeds 0.2, aligning with observations in prior works.
> Collectively, these results confirm QHP’s robustness to variations in the temperature parameter and eliminate the need for tedious fine-tuning within standard ranges, thereby enhancing its practicality.
>
>
> References
>
> [1] Liang Wang, Nan Yang, Xiaolong Huang, Linjun Yang, Rangan Majumder, and Furu Wei. Improving text embeddings with large language models. In Proceedings of the 62nd Annual Meeting of the Association for Computational Linguistics, pp. 11897–11916, 2024.
>
> [2] Ting Chen, Simon Kornblith, Mohammad Norouzi, and Geoffrey E. Hinton. A simple framework for contrastive learning of visual representations. In International conference on machine learning (ICML), volume 119, pp. 1597–1607, 2020.

---

### Author Response · Authors · 2025-11-24
**To PC, SAC , AC, and all Reviewers**

To PC, SAC , AC, and all Reviewers:

We sincerely appreciate the time and effort the PC, SAC, AC, and all reviewers have dedicated to reviewing our work. We are grateful for the detailed and thoughtful feedback on our submission, particularly the positive comments and insights. Below, we summarize the strengths recognized by the reviewers:

1.**Novelty**: Introducing hubness into FS-3Dseg is novel and offers a fresh perspective on addressing prototype bias. (Reviewer SJ47, Reviewer F9pR, Reviewer 63hN, Reviewer B2Tj)

2.**Effectiveness**: Comprehensive experiments on S3DIS and ScanNet, supported by ablations and sensitivity analyses, demonstrate the effectiveness of the proposed QHP framework. (Reviewer SJ47, Reviewer F9pR, Reviewer 63hN)

3.**Readable**: The paper is clearly written, easy to follow, and supported by helpful figures. (Reviewer F9pR, Reviewer B2Tj)

4.**Performance**: The method achieves strong and consistent improvements over baselines, with stepwise gains from HPG and PDO. (Reviewer SJ47, Reviewer 63hN, Reviewer B2Tj)

Thank you once again for your time and effort in reviewing this manuscript. For the questions and other concerns raised by the reviewer, we respond to each in detail and outlined our plans for improvement. If any questions are not answered or our response is unclear, we would appreciate the opportunity to communicate further with our reviewer.

---

### Author Response · Authors · 2025-12-03
**Summary of Contributions and Rebuttal**

Dear PCs, SACS, ACs,

We would like to express our sincere appreciation for the reviewers' thoughtful and constructive feedback. In this document, we summarize the reviews and detail the revisions and clarifications we have made in response to the concerns raised.

---

## Claim of Contribution

In this paper, We present **Query-aware Hub Prototype (QHP) Learning**, a framework that reduces prototype bias in few-shot 3D point cloud segmentation by aligning prototype generation with query semantics. QHP first identifies query-relevant support hubs through bipartite hubness analysis and constructs prototypes that better capture cross-set semantic structure. To further refine ambiguous or low-purity prototypes, a purity-reweighted contrastive optimization pulls misaligned hubs toward their class centers. Together, these components significantly improve prototype–query consistency and yield strong performance gains on S3DIS and ScanNet benchmarks.

---

## Summary of Reviews and Responses

In the current review, we are grateful for the positive and constructive feedback from the reviewers. Reviewer SJ47 highlights **the novelty of integrating hubness into FS-3DSeg**, addressing prototype bias. Reviewer F9pR emphasizes the **comprehensive empirical validation** via extensive experiments and ablations. Reviewer 63hN appreciates **the overall effectiveness**, particularly improvements from HPG and PDO. Reviewer B2Tj commends clarity and organization, noting the figures enhance readability.

We now address the specific concerns raised by the reviewers, providing clarifications and additional results as outlined below.

|Reviewer | Reviewer's Concern / Questions| Author's Response  |
| ------ | -- | --- |
| SJ47, 63hN | Hyperparameter sensitivity and the additional temperature parameter $\tau$ ablation | Extensive ablations (Tables 15–18 in Appendix A.4) on ScanNet demonstrate robust results across variations in the hub ratio, HPG ($k,\eta$), PDO ($k,\gamma$), and $\lambda$. The results show strong cross-dataset consistency and no need for re-tuning. Ablations on temperature parameter $\tau$ (Table 11, Appendix A.2) reveal stability within the range $\tau=0.05–0.2$. |
| F9pR | Comparison with Recent Baselines | We clarify that many recent works are not directly comparable due to differences in modalities, tasks, or data availability (closed-source, etc.). A preliminary comparison with MM-FSS (a recent multi-modal FS-3DSeg method using 2D + text supervision) under the same baseline shows that QHP remains competitive despite using only point-cloud supervision. |
| F9pR | Robustness under domain shift | We present cross-domain evaluations (S3DIS → ScanNet) in Table 20 (Appendix A.6), where QHP achieves +1.95 mIoU over the baseline. |
| F9Pr | Efficiency and scalability | We provide a detailed breakdown of FLOPs and inference time for HPG/PDO in Table 14 (Appendix A.3). Notably, graph construction and clustering account for only 6–12% of the total FLOPs, with the backbone dominating the computational load. |
| F9pR, B2Tj | Need for qualitative visualization and clearer motivation | We have included failure-case visualizations in Figure 9 (Appendix A.5) and added multi-stage t-SNE comparisons in Figure 6 (Sec. 4.3). These visualizations demonstrate COSeg prototype drift, HPG center alignment, and PC-loss boundary refinement, providing clearer insight into the effectiveness of our approach. |
| 63hN | Novelty vs. existing query-guided methods | We clarify that prior works integrate query information only after prototype formation, whereas QHP integrates query–support interaction during prototype generation. We have updated the motivation section (Lines 15–18 and 52–72) to reflect this distinction more clearly. |
| 63hN | PC loss vs. standard contrastive loss | We provide a full \(\lambda\)-sweep comparison in Table 12 (Appendix A.2), which demonstrates that PC loss consistently outperforms standard contrastive loss. Additionally, clearer prototype boundaries are validated through t-SNE visualizations in Figure 6 (c–d) (Sec. 4.3) and bad-hub statistics in Figure 7 (Appendix A.4). |
| 63hN | Why pull bad hubs instead of reinforcing good hubs  | Theoretical reasoning for our choice to correct bad hubs rather than reinforce good hubs is discussed in Sec. 3.4. Empirical evidence (Table 10, Appendix A.2) shows that the "pull-good-hubs" strategy results in a 2.5 mIoU reduction compared to QHP’s bad-hub correction approach. |
| B2Tj | Module rationale (HPG vs PDO) | We clarify that HPG uses query-only centers (Lines 233–235) to ensure query relevance, while PDO utilizes both query and support centers (Lines 254–256) to detect global bad hubs. |

We believe these clarifications address the reviewers' concerns and strengthen our contributions. We appreciate the opportunity to engage in this constructive discussion.

---

Best regards,

Authors of submission 8774

---

### Meta-Review · Area_Chair_an2d · 2026-01-04

**Summary:**

Four reviewers assessed this submission, yielding a split decision with three reviewers leaning towards borderline rejection and one recommending acceptance. The primary concerns initially raised by the panel focused on the method's generalization capabilities, specifically the sensitivity of the "Hub Point Mining" and QHP modules to hyperparameters across different datasets like S3DIS and ScanNet. Reviewers also criticized the limited comparisons to recent transformer-based or query-guided baselines, the marginal magnitude of performance improvements, and the validity of the authors' claims regarding the novelty of using query data for prototype generation. During the rebuttal, the authors provided additional experiments, including efficiency analyses, failure case visualizations, and clarifications on technical inconsistencies. While the rebuttal addressed some of the concerns regarding efficiency and implementation details, doubts remain regarding the robustness of the method under significant domain shifts and the precision of the paper's positioning against prior art. Consequently, AC recommends rejection.

**Reviewer Concerns:**

The authors successfully addressed several specific critiques, including the request for efficiency analyses (Reviewer F9pR), the clarification of weighting schemes and mining definitions (Reviewer B2Tj), and the inclusion of missing temperature ablations (Reviewer SJ47). However, critical issues remain unresolved. The concern raised by Reviewer 63hN regarding the manuscript's inaccurate claim, that prior metric-based methods do not leverage query data, was not adequately corrected. Doubts regarding generalization persist: Reviewer SJ47 noted that the "Query-Guided Hub Point" (QHP) module remains sensitive to hyperparameters despite new ScanNet experiments, and Reviewer F9pR observed that performance gains in simulated domain-shift scenarios were marginal. This combination of hyperparameter fragility and incremental improvement suggests the method lacks the robustness required for a higher tier of acceptance.

**Reviewer Scores:**

Reviewer SJ47 is expected to maintain his/her score of 4. While the authors provided the requested ablation studies, the results confirmed the hyperparameter sensitivity of the QHP module rather than resolving the generalization concern, leaving the reviewer's primary hesitation valid. Reviewer F9pR will likely also retain a score of 4; although the authors addressed inquiries regarding efficiency and failure cases, the supplementary experiments revealed only marginal improvements in cross-domain scenarios, reinforcing the reviewer's skepticism about the method's real-world robustness. Reviewer 63hN is expected to keep his/her score of 4, as the fundamental issue regarding the overclaimed novelty and the inaccurate characterization of prior query-guided methods was not satisfactorily resolved. Reviewer B2Tj will likely maintain his/her score of 8, as the rebuttal fully addressed their technical questions regarding consistency and related work.

---

### Decision · Program_Chairs · 2026-01-26

Reject